# Recovering Private Text in Federated Learning of Language Models

**Samyak Gupta**[*]
Princeton University
samyakg@cs.princeton.edu

**Yangsibo Huang**[*]
Princeton University
yangsibo@princeton.edu

**Zexuan Zhong**
Princeton University
zzhong@cs.princeton.edu

**Tianyu Gao**
Princeton University
tianyug@cs.princeton.edu

**Kai Li**
Princeton University
li@cs.princeton.edu

**Danqi Chen**
Princeton University
danqic@cs.princeton.edu

## Abstract

Federated learning allows distributed users to collaboratively train a model while keeping each user's data private. Recently, a growing body of work has demonstrated that an eavesdropping attacker can effectively recover image data from gradients transmitted during federated learning. However, little progress has been made in recovering text data. In this paper, we present a novel attack method FILM for federated learning of language models (LMs). For the first time, we show the feasibility of recovering text from *large batch sizes* of up to 128 sentences. Unlike image-recovery methods that are optimized to match gradients, we take a distinct approach that first identifies a set of words from gradients and then directly reconstructs sentences based on beam search and a prior-based reordering strategy. We conduct the FILM attack on several large-scale datasets and show that it can successfully reconstruct single sentences with high fidelity for large batch sizes and even multiple sentences if applied iteratively. We evaluate three defense methods: gradient pruning, DPSGD, and a simple approach to freeze word embeddings that we propose. We show that both gradient pruning and DPSGD lead to a significant drop in utility. However, if we fine-tune a public pre-trained LM on private text without updating word embeddings, it can effectively defend the attack with minimal data utility loss. Together, we hope that our results can encourage the community to rethink the privacy concerns of LM training and its standard practices in the future.[2]

## 1 Introduction

Federated learning (McMahan et al., 2017) is a method to allow multiple participants to collaboratively train a global model without exchanging their private data. At each step of training, a central server transmits model parameters to every participating client. Each client then computes a model update (i.e., gradients) using its local data and sends it to the server. Finally, the server aggregates all updates—typically by averaging them—and updates the model. Federated learning is actively being considered for privacy-sensitive applications such as virtual mobile keyboards and analysis of electronic health records in hospitals (Li et al., 2020a).

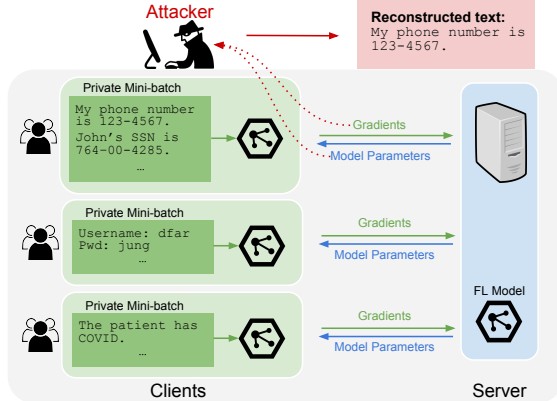

Figure 1: FILM allows an *honest-but-curious* eavesdropper on the communication between the client and the server in federated learning to recover private text of the client.

---

[*]The first two authors contributed equally.

[2]Our code is publicly available at https://github.com/Princeton-SysML/FILM.

36th Conference on Neural Information Processing Systems (NeurIPS 2022).

However, recent studies (Zhu et al., 2019; Geiping et al., 2020; Pustozerova & Mayer, 2020; Lyu et al., 2020; Yin et al., 2021; Zhu & Blaschko, 2021; Huang et al., 2021; Gupta et al., 2021) show that an *honest-but-curious* eavesdropper in federated learning can recover private data of the client (Fig. 1). However, success in recovering text data remains limited to unrealistically small batch sizes (see Sec. 2 for more details).

In this work, we study federated learning of neural language models (LMs) and present a novel attack called FILM (**F**ederated **I**nversion Attack for **L**anguage **M**odels), which can recover private text data from information transmitted during training. For the first time, we demonstrate that an attacker can successfully recover sentences from *large* training batches of up to 128 sentences, making it practical and alarming in real-world scenarios. We focus on language models for two reasons: (1) Transformer-based (Vaswani et al., 2017) language models have become the backbone of modern NLP systems (Devlin et al., 2019; Radford et al., 2018, 2019; Brown et al., 2020) and are quickly adapted to a number of privacy-sensitive domains (Alsentzer et al., 2019; Huang et al., 2019; Li et al., 2020b; Liu & Miller, 2020; Kraljevic et al., 2021); (2) We identify that training LMs is at a higher risk of information leakage by nature because the attacker can possibly leverage the memorization capability of LMs during federated learning.

Our FILM attack deviates from previous gradient-matching approaches for image data that directly optimize in high-dimensional, continuous space, which we find to be ineffective for discrete text inputs and highly sensitive to initializations (Sec. 6). Instead, we first recover a set of words from the word embeddings' gradients (Melis et al., 2019), and this step provides a set of words that appear in any sentences of the training batch without the frequency information[3]. We then develop a simple and effective strategy using beam search (Reddy et al., 1977; Russell & Norvig, 2010), which attempts to reconstruct one sentence from the set of words. This step is crucial as it takes advantage of prior knowledge encoded in pre-trained language models and memorization of training data during federated training. We also further design a token-reordering approach that leverages both language prior and gradient information to refine the recovered sentence. Finally, we show that we can recover multiple sentences from the same training batch by iteratively applying the same procedure.

We evaluate FILM on two language modeling datasets: WikiText-103 (Merity et al., 2017) and Enron Email (Klimt & Yang, 2004), based on a GPT-2 model (Radford et al., 2019), using either pre-trained or randomly-initialized weights. We analyze the attack performance with different batch sizes, the number of training data points, and the number of training epochs. Our experiments demonstrate high-fidelity recovery of a single sentence feasible, and recovery of significant portions of sentences for training batches of up to 128 sentences. Furthermore, we find that recovery of multiple sentences from the same batch (1/3 for a batch size of 16) is possible through repeated applications of FILM.

For defense, we first evaluate two previously proposed defense methods, gradient pruning (Zhu et al., 2019) and Differentially Private Stochastic Gradient Descent (DPSGD) (Abadi et al., 2016) against the proposed attack and find that both of them suffer substantial utility loss. We then propose a simple method to *freeze* the word embeddings of the model during training to prevent the critical first stage of our attack and find that (1) if we train an LM from the scratch on the private text, it will cause more utility loss; (2) however, if we start from a public LM (e.g., pre-trained GPT-2), it can effectively defend the FILM attack with minimal drop in utility. Together, our work presents a strong attack on federated learning of LM training, and raises privacy concerns in current practices. We hope our new (and simple) proposal of *freezing* word embeddings from a public LM encourages reconsideration of standard practices for training LMs on privacy-sensitive domains.

## 2 Related work

**Data recovery from gradients.**    Gradients in federated learning have been shown to leak information of private data. Prior works reconstruct private training images from gradients by treating the reconstruction as an optimization problem (Zhu et al., 2019; Zhao et al., 2020; Geiping et al., 2020; Enthoven & Al-Ars, 2021; Yin et al., 2021; Zhu & Blaschko, 2021; Jin et al., 2021; Jeon et al., 2021). They iteratively manipulate reconstructed images to yield similar gradients as observed gradients computed on the private images, using regularization terms based on image priors such as total variation (Zhu et al., 2019; Geiping et al., 2020) and batch normalization statistics (Yin et al., 2021).

---

[3]We may still estimate the frequency of words based on the magnitude of gradients (Wainakh et al., 2021; Fowl et al., 2022). However, later we show that our attack can actually compensate for the lack of frequency information (Sec. 4.3).

The first attempt to recover text from gradients is Zhu et al. (2019), which briefly presented leakage results in masked language modeling (Devlin et al., 2019). This attack matches target gradients with continuous representations and maps them back to words that are closest in the embedding matrix. Deng et al. (2021) improve the method by adding a regularization term that prioritizes gradient matching in layers closer to the input data. Both approaches only work with a batch size of 1 (Sec. 6). Recently, Dimitrov et al. (2022) propose a method to minimize the loss of gradient matching and the probability of prior text computed by an auxiliary language model and show the recovery of a batch of 4 sentences with binary classification tasks with a BERT model. In addition, Boenisch et al. (2021); Fowl et al. (2022) recover text data in federated learning under a stronger threat model: the server is malicious and can manipulate the training model's weights to enable easier text reconstruction.

Our work builds on the insight from Melis et al. (2019) that show the information leakage from the embedding layer by observing non-zero entries in the gradients of the word embedding matrix. Different from their attack which only infers a person or location from a shallow GRU model (Cho et al., 2014), we propose a much stronger attack that aims at recovering full sentences from larger language models. We include a high-level comparison of key differences between our approach and prior approaches in Table 6 in the appendix.

**Memorization in language models.** Our method is also inspired by the work related to memorization of training data in language models. Recent studies (Carlini et al., 2019; Thakkar et al., 2021; Song & Raghunathan, 2020; Zanella-Béguelin et al., 2020; Carlini et al., 2021; Bender et al., 2021) notice that large language models can memorize their training dataset and thus can be prompted to output specific sensitive information. While these previously demonstrated memorization attacks try to recover *a small subset of* sensitive text from a large training corpora, our attack aims at recovering sentences from *every* mini-batch in federated learning. We stress that our attack is significantly more powerful than the previous works studying memorization.

**Defenses against gradient inversion attacks for text data.** Cryptographic approaches such as encrypting gradients (Bonawitz et al., 2016) or encrypting the data and model (Phong et al., 2018) can guarantee secure training in a federated learning setting against gradient inversion attacks. However, practical deployment of these approaches often slows down model training and requires special setup. A few efficient methods are proposed to defend gradient inversion attacks are for text data. Zhu et al. (2019) mention briefly adding differentially private noise to the gradients (Abadi et al., 2016), or setting gradients of small magnitudes to zero (i.e., gradient pruning), but these defenses typically hurt the accuracy of the trained models (Li et al., 2021; Yu et al., 2021). Huang et al. (2020b) propose InstaHide that encodes each input training image to the neural network using a random pixel-wise mask and the MixUp data augmentation (Zhang et al., 2018). The idea is later extended to language understanding tasks (Huang et al., 2020a): instead of 'hiding' the input word embeddings, they perform MixUp on [CLS] tokens of BERT models.

## 3 Preliminaries

In this section, we describe relevant background of language models, federated learning, as well as the threat model of our proposed attack method.

### 3.1 Language Modeling

Language modeling is a core task in natural language processing and has been used as a building block of state-of-the-art pipelines (Radford et al., 2018; Brown et al., 2020). Given a sequence of $n$ tokens $\mathbf{x} = \{x_1, x_2, ..., x_n\}$, the language modeling task is to estimate the probability distribution of $\mathbb{P}(\mathbf{x})$:

$$\log \mathbb{P}_\theta(\mathbf{x}) = \sum_{i=1}^{n} \log \mathbb{P}_\theta(x_i | x_1, ..., x_{i-1}). \tag{1}$$

Modern neural language models are parameterized by RNNs (Mikolov et al., 2010) or Transformers (Vaswani et al., 2017), which consist of millions or billions of parameters (denoted $\theta$). A language model first maps $\mathbf{x}$ to a sequence of vectors via a word embedding matrix $\mathbf{W} \in \mathbb{R}^{|\mathcal{V}| \times d}$, where $\mathcal{V}$ is the vocabulary and $d$ is the hidden dimension. After computing the hidden representation $\mathbf{h}_i$ conditioned on $x_1, \ldots, x_{i-1}$, the model predicts the probability of the next token as:

$$\mathbb{P}_\theta(x_i | x_1, ..., x_{i-1}) = \frac{\exp(\mathbf{h}_i^\top \mathbf{W}_{x_i})}{\sum_{j \in \mathcal{V}} \exp(\mathbf{h}_i^\top \mathbf{W}_j)}. \tag{2}$$

### 3.2 Federated Learning

Federated learning (FL) (McMahan et al., 2017) is a communication protocol for training a shared machine learning model on decentralized data. An FL system usually involves $N$ clients: $c_1, c_2, \cdots, c_N$

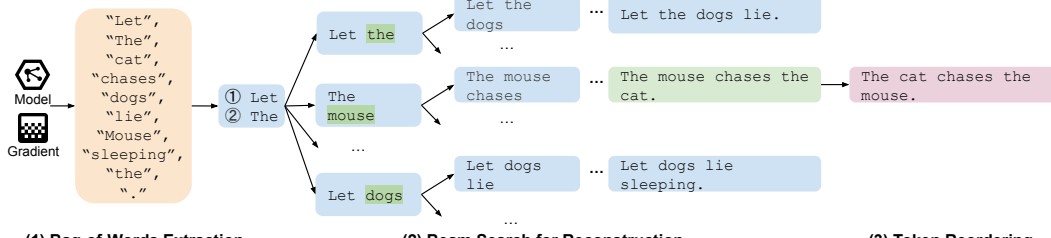

**(1) Bag-of-Words Extraction**    **(2) Beam Search for Reconstruction**    **(3) Token Reordering**

Figure 2: Illustration of our pipeline for a batch size of 2. We first recover a bag of words from the observed gradients, and then generate candidate sentences using beam search from the set of words, retaining top beams at every step according to a beam scoring function. Finally, we re-order phrases and words in the best candidate.

and a central server $s$, where the clients wish to collaboratively train a neural network $f_\theta$ with their private data $\mathcal{D}_1, \mathcal{D}_2, \cdots, \mathcal{D}_N$, under the coordination of the server. The training process optimizes $\theta$ (model parameters) using a loss function $\mathcal{L}$ and runs for $T$ iterations. This work focuses on federated learning of language models, where clients can start from a randomly-initialized neural network, or a pre-trained public model (Hard et al., 2018). At each iteration $t \in [T]$, an individual client $c_i, i \in [N]$ computes $\nabla_{\theta^t} \mathcal{L}_{\theta^t}(\mathcal{B}_i)$, the gradients of the current model parameters $\theta^t$ on a randomly-sampled private data batch $\mathcal{B}_i \subset \mathcal{D}_i, |\mathcal{B}_i| = b$, and shares the gradients with the server $s$. The server then aggregates gradient updates from all clients and updates the model:

$$\theta^{t+1} = \theta^t - \eta \sum_{i=1}^{N} \nabla_{\theta^t} \mathcal{L}_{\theta^t}(\mathcal{B}_i), \tag{3}$$

where $\eta$ is the learning rate. Finally, the updated parameters $\theta^{t+1}$ are broadcast to individual clients.[4]

### 3.3 Threat Model

**Adversary's capabilities.**    We consider an honest-but-curious adversary who is eavesdropping on the communication between the central server $s$ and an arbitrary client $c_i, i \in [N]$ in the federated training of a language model parameterized by $\theta$, as described above. An adversary in this scenario has white-box access to 1) the gradients $\nabla_{\theta^t} \mathcal{L}_{\theta^t}(\mathcal{B}_i)$ sent by the client $c_i$, and 2) the model parameters $\theta^t$ (Fig. 1), including the vocabulary $\mathcal{V}$ and the embedding matrix $\mathbf{W}$. Note that the adversary can inspect these information at *any* training iterations $t \in [T]$.

**Adversary's objective.**    The adversary aims to recover *at least one* sentence from the private training data batch $\mathcal{B}_i$ based on the information they observe, as recovering a single sentence is already sufficient to break the privacy guarantee of federated learning. Since each training batch consists of randomly selected sentences from the private training dataset, if the adversary could recover at least one sentence from each training batch during the whole training procedure, they would be able to recover a considerable fraction of the original training dataset. Additionally, the attacker may also apply the attack on a single batch multiple times to recover more sentences from the batch (Sec. 4.5). The strength of an attack will be measured by the similarity between the recovered sentence and its corresponding private sentence in the original batch.

## 4 Proposed Attack: FILM

### 4.1 Overview

In this section, we present a novel attack method FILM, which can construct sentences from *any* training batches ($|\mathcal{B}_i| \leq 128$). Our method consists of three steps:

(1) Inspired by Melis et al. (2019), we first recover a set of words that may appear in a private batch ($\mathcal{B}_i$) from gradients (Sec. 4.2). While their goal is to identify individual words from gradients, our approach is able to recover the order of words in a sentence.

(2) We attempt searching for full sentences using beam search using the set of words, based on the probability distribution $\mathbb{P}_{\theta^t}(x)$, which leverages either the language prior from a pre-trained language model ($t = 0$), or the memorization capability of the LM during training ($t > 0$) (Sec. 4.3).[5]

(3) Finally, we design a prior-based scoring function and demonstrate that we can further improve the quality of recovered sentences by performing a simple and lightweight reordering step (Sec. 4.4).

---

[4]Note that Eq. 3 uses SGD for an easier demonstration for an update step in federated learning. People usually use Adam (Kingma & Ba, 2015) in real-world training of neural language models.

[5]Recall that the attacker can inspect the gradients at any step $t \in [T]$ during the $T$-epoch training. An attacker performing our attack at a larger $t$ may have much stronger results than for a smaller $t$, as we later show in Sec. 7.

We show that the above pipeline is capable of recovering single sentences from large training batches successfully. In Sec. 4.5, we discuss that we can iterate the above procedure to recover multiple sentences from the same training batch, which makes our attack even stronger. In the following, we describe our attack method in detail. To better illustrate our attack's functionality, we provide a simplified step-by-step example that recovers a batch of 2 simple sentences in Fig. 2.

## 4.2 Bag-of-Words Extraction

**Input:** $\nabla_{\theta^t} \mathcal{L}_{\theta^t}(\mathcal{B}_i)$, gradients of the language model on the private batch of sentences.

**Output:** bag of words $B$: {"Let", "The", "cat", "chases", "dogs", "lie", "mouse", "sleeping", "the", "."}. Our first step is to extract a bag of words from $\nabla_{\theta^t} \mathcal{L}_{\theta^t}(\mathcal{B}_i)$, the gradients observed by an eavesdropper at the $t$-th iteration of training. We follow the method outlined by Melis et al. (2019) and recover a set of words $B = \{w_1, w_2, \dots, w_m\}$ by considering the non-zero rows of the word embedding gradients $\nabla_{\mathbf{W}^t} \mathcal{L}_{\theta^t}(\mathcal{B}_i)$. Additionally, we are also able to recover the maximum length of sentences in the batch by analyzing the non-zero rows of position embeddings gradients $\nabla_{\mathbf{P}^t} \mathcal{L}_{\theta^t}(\mathcal{B}_i)$.[6] Note that this step only determines the set of words but not their *frequency*. In practice, we find that augmenting our beam search with an n-gram penalty (Sec. 4.3) can compensate for this lack of information.

## 4.3 Beam Search for Sentence Reconstruction

**Input:** $B$, the bag of private words recovered by Sec. 4.2.
**Output:** a recovered sentence from the batch, "The mouse chases the cat.".

The second stage of our attack is to recover *one* sentence based on the extracted bag of words $B$ and the learned probability distribution of text $\mathbb{P}_{\theta^t}(\mathbf{x})$ at training step $t$. Since it is intractable to generate text directly using $\mathbb{P}_{\theta^t}(\mathbf{x})$, we use the beam search (Reddy et al., 1977; Russell & Norvig, 2010), which is a greedy auto-regressive algorithm to generate sequences in a step-by-step manner with a trained language model. We provide a detailed algorithm in Appendix A.

To initiate the beam search, we first find a set of words that can be chosen as starting tokens of sentences. A straightforward method is to select words that begin with a capital letter from $B$. For example, we would select "Let" and "My" as possible prompts for the illustrated batch. Then at each beam search step, we keep the top $k$ possible beam states, find the $k$ most likely next words to each beam by using the language model (see procedure TOP($\cdot$) in Appendix A, Algorithm 1), and keeps the top $k$ new beam states. The beam tree continues until the generated sentence reaches a specified length (we use 40 words), and outputs the text that has the overall highest probability (see Fig. 2).

As the bag of words $B$ does not contain information on the frequency of each word in the sentence, and beam search is prone to generating repetitive content (Holtzman et al., 2020), we also include a penalty for repeated $n$-grams (Vijayakumar et al., 2016) in a sentence (see line 3 in procedure TOP($\cdot$)). Noticeably, we compare the performance of our attack with an oracle scenario where the attacker knows the frequency of each word in Fig. 5, and find that adding an n-gram penalty ($n = 2$) can achieve performance comparable without frequency information.

## 4.4 Prior-Guided Token Reordering

**Input:** a recovered sentence from Sec. 4.3, "The mouse chases the cat."
**Output:** a reordered sentence, "The cat chases the mouse."

By construction, beam search is a greedy method that generates sentences from left to right. Despite that it generally produces fluent sentences by maximizing $\mathbb{P}_{\theta^t}(\mathbf{x})$, it does not leverage the *gradients* information which is the main signal in previous image attack methods. We also notice that beam search may recover the general structure of the sentence, while failing to recover the order of specific phrases or words. Thus, the final stage of our attack is to reorder tokens in the extracted sentence from the previous step, with the objective to improve its quality measured by a prior-based scoring function. The scoring function leverages both the perplexity and the gradient norm. Formally, given a sentence $\mathbf{x}$, its prior score $\mathcal{S}_\theta(\mathbf{x})$ is defined as:

$$\mathcal{S}_\theta(\mathbf{x}) = \underbrace{\exp\left\{-\frac{1}{n}\log\mathbb{P}_\theta(\mathbf{x})\right\}}_{\text{Perplexity}} + \beta\underbrace{\|\nabla_\theta\mathcal{L}_\theta(\mathbf{x})\|}_{\text{Gradient Norm}}, \tag{4}$$

where $\log\mathbb{P}_\theta(\mathbf{x})$ is the log-likelihood of the sentence (Eq. 1) according to the model parameters $\theta$, and $\beta$ is a hyperparameter that controls the importance of gradient norm. Appendix B.8 provides empirical analysis that supports the design of this scoring function. We conduct two additional steps to reorder both phrases and tokens:

---

[6]We assume that sentences in one batch vary in length. A standard practice is to pad shorter sentences.

**Phrase-wise reordering.** We first preprocess the sentence $\mathbf{x}$ by removing its redundant segments[7]. We then adjust the preprocessed sentence by reordering the phrases inside it. At each iteration, we generate multiple candidates by first cutting the sentence at $p$ positions using the strategy in Malkin et al. (2021), and then permuting the phrases between the cuts to form a new sentence. We then select the best candidate measured by $\mathcal{S}_\theta(\cdot)$. This stage terminates after 200 steps.

**Token-wise reordering.** We then reorder individual tokens. At each reordering step, we generate candidates using the following token-wise operations: 1) randomly swaps two tokens in the sentence, or 2) randomly deletes a token from the sentence, or 3) randomly inserts a token from the bag of words (recovered by Sec. 4.2) into the sentence. We then select the best candidate measured by $\mathcal{S}(\cdot)$ and move to the next iteration. This stage terminates after 200 steps.

### 4.5 Recovering Multiple Sentences

The previous steps only consider the recovery of a single sentence from a training batch. In this step, we describe an extension of our attack which is able to support recovery of multiple sentences from the same batch. We first perform beam search as described in Sec. 4.3. We store the results and repeat the beam search again, except this time applying an additional $n$-gram penalty based on the results of the previous search. Over many repetitions of this procedure, we are able to build a set of candidate recoveries for the original batch. Finally, we use the reordering method described in Sec 4.4 to improve each candidate.

## 5 Defending Against the FILM Attack

We evaluate previously proposed defense methods, gradient pruning (Zhu et al., 2019) and DPSGD (Abadi et al., 2016) and find that both are less effective in defending our attack (Sec. 6.3).

To defend against the FILM attack, we propose a defense method which simply *freezes the word embeddings* of the model during training. Our key insight is to focus on preventing the first step of the attack, i.e., the recovery of the bag-of-words from word embedding gradients. If the attack fails at this step, the later stages of the attack are no better than beam search over the entire vocabulary as with the prior attack (Carlini et al., 2021). Similarly, previously proposed gradient inversion attacks (Zhu et al., 2019; Fowl et al., 2022) for smaller batch sizes also depend on this step. By freezing the word embeddings during training, it is able to completely prevent the recovery of bag-of-words. In other words, once we prevent the transmission of gradients of word embeddings during training, it is no longer possible to recover the bag-of-words.

We consider the settings for both 1) training an LM from the scratch; 2) continuing training from a public, pre-trained LM (e.g., GPT-2) on the private text, when freezing the word embeddings. As we later show in Sec. 6.3, the first setting causes more utility loss (because the word embeddings are randomly initialized) while the latter setting provides the best privacy-utility trade-off. Since updating word embeddings is a common practice in training LMs, we suggest researchers and practitioners consider training from a public LM and freezing word embeddings in privacy-sensitive applications.

## 6 Experiments

### 6.1 Setup

**Model and datasets.** We evaluate the proposed attack with the GPT-2 base (117M parameters) model (Radford et al., 2019) on two language modeling datasets, including WikiText-103 (Merity et al., 2017) and the Enron Email dataset (Klimt & Yang, 2004). Both datasets are publicly available for research uses. We choose WikiText-103 because it is commonly used in language modeling research, and Enron Email because it consists of private email messages that contain abundant private information such as individuals' names, addresses, and even passwords. Note that the GPT-2 model is not trained on any Wikipedia data, and very unlikely on the Enron Email data. We evaluate our attack on a subset of WikiText-103 and Enron Email. After preprocessing (more details provided in Appendix B), we have 203,456 sentences from the WikiText-103 dataset and 31,797 sentences from the Enron Email dataset.

**Training and attack settings.** Following previous studies (Zhu et al., 2019; Geiping et al., 2020; Yin et al., 2021; Huang et al., 2021) for gradient inversion attacks, our evaluation considers a federated

---

[7]We notice that the sentences we recover from the previous step may not naturally finish after the beam search generates a punctuation. To address this problem, we preprocess each recovered sentence $\mathbf{x}$ by removing segments after its first punctuation, as long as the resulting sentence $\mathbf{x}'$ satisfies $\mathcal{S}_\theta(\mathbf{x}') < \mathcal{S}_\theta(\mathbf{x})$.

| Attack & Batch Size $b$ | Original Sentence | Best Recovered Sentence |
|---|---|---|
| **WikiText-103** | | |
| Zhu et al. (2019), $b = 1$ | As Elizabeth Hoiem explains, "The most English of all Englishmen, then, is both king and slave, in many ways indistinguishable from Stephen Black. | thelesshovahrued theAsWords reporting the Youngerselagebalance, mathemat mathemat mathemat reper arrangpmwikiIndia Bowen perspectoulos subur, maximal |
| Deng et al. (2021), $b = 1$ | Both teams recorded seven penalties, but Michigan recorded more penalty yards. | Both recorded teams seven but Michigan to recorded penalties 40 more penalty outstanding |
| FILM, $b = 1$ | The short@-@tail stingray forages for food both during the day and at night. | The short@-@tail stingray forages for food both during the day and at night. |
| FILM, $b = 16$ | A tropical wave organized into a distinct area of disturbed weather just south of the Mexican port of Manzanillo, Colima, on August 22 and gradually moved to the northwest. | Early on September 22, an area of disturbed weather organized into a tropical wave, which moved to the northwest of the area, and then moved into the north and south@-@to the northeast. |
| FILM, $b = 128$ | A remastered version of the game will be released on PlayStation 4, Xbox One and PC alongside Call of Duty: Infinite Warfare on November 4, 2016. | At the time of writing, the game has been released on PlayStation 4, Xbox One, PlayStation 3, and PC, with the PC version being released in North America on November 18th, 2014. |
| **Enron Email** | | |
| Zhu et al. (2019), $b = 1$ | Don and rogers have decided for cost management purposes to leave it consolidated at this point. | dj "... Free, expShopcriptynt)beccagressive Highlands andinos Andrea Rebell impacts |
| Deng et al. (2021), $b = 1$ | We should not transfer any funds from tenaska iv to ena. | We should transfer any funds not from tenaska iv to iv en happens |
| FILM, $b = 1$ | Volume mgmt is trying to clear up these issues. | Volume mgmt is trying to clear up these issues. |
| FILM, $b = 16$ | Yesterday, enron ousted chief financial officer andrew fastow amid a securities and exchange commission inquiry into partnerships he ran that cost the largest energy trader $35 million. | Yesterday, enron ousted its chief financial officer, andrew fastow, amid a securities and exchange commission inquiry into partnerships he ran that cost the company $35 million in stock and other financial assets. |
| FILM, $b = 128$ | Yesterday, enron ousted chief financial officer andrew fastow amid a securities and exchange commission inquiry into partnerships he ran that cost the largest energy trader $35 million. | Yesterday, enron ousted chief financial officer andrew fastow amid a securities and exchange commission inquiry into partnerships he ran that he said cost the company more than $1 billion in stock and other assets. |

Table 1: Performance of Zhu et al. (2019)'s and Deng et al. (2021)'s with batch size = 1 (more in Appendix B.3) and FILM (ours) with different batch sizes. We show the best recovered sentence among 20 tested batches for each batch size (see Fig. 3 for average results). Text in green represents successfully recovered phrases and words.

learning setting with a single server and a single client.[8] Unless otherwise noted, we train the model on these sentences for $90,000$ iterations using an initial learning rate of $1 \times 10^{-5}$, with a linearly decayed learning rate scheduler. All models were trained using early stopping, i.e., models were trained until the loss of the model on the evaluation set increased. We note that the running time of our algorithm is quite fast, and we can recover a single sentence in under a minute using an Nvidia 2080TI GPU.

**Evaluation metrics.** We use the following metrics to evaluate the attack performance: (a) **ROUGE** (Lin, 2004) is a set of metrics for evaluating summarization of texts as well as machine translation. We use ROUGE-1/2/L F-Scores to evaluate the similarity between the recovered and the original sentences, following (Deng et al., 2021; Dimitrov et al., 2022). More specifically, ROUGE-1/2 refer to the overlap of unigrams and bigrams between the recovered and the original text respectively. ROUGE-L measures the longest matching subsequence. For ablation studies we only show ROUGE-L as it is more representative of significant leakage than ROUGE-1 or ROUGE-2. (b) We also propose to use **named entity recovery ratio (NERR)** as the percentage of original named entities that can be perfectly recovered. Since named entities usually contain sensitive information (e.g., names, addresses, dates, or events), NERR measures how well the attacker recovers such information, ranging from a complete mismatch (NERR = 0) to a perfect recovery (NERR = 1).

## 6.2 Performance of Our Attack

**Scaling to large training batches.** We first compare the performance with respect to different batch sizes. In this setting, we start from a pre-trained GPT-2 model and fine-tune it on both datasets respectively (WikiText-103 and Enron Email). Fig. 3 shows the performance of our attack using different batch sizes with the WikiText-103 dataset and the Enron Email dataset. We observe that for a batch size of 1, all four scores (ROUGE-1, ROUGE-2, ROUGE-L, NERR) are close to 1, indicating a near-perfect recovery. We observe that as the batch size increases, the quality of the recovery by ROUGE-1, ROUGE-2, and NERR start to decrease. We attribute this drop in performance to the bigger set of words in the same training batch, as well as the growing search space of possible word orderings. We also observe that the Enron Email dataset is more susceptible to attack.

Table 1 shows that the approach of Zhu et al. (2019) fails to recover the original sentence for batch size $b = 1$. We find that the gradient matching method in Zhu et al. (2019) is sensitive to initialization and it can perform better recovery when using a good initialization (see Appendix B.3). Deng et al. (2021)

---

[8]This setting is often accepted as an adequate stand-in for federated learning as synchronous federated learning with $N$ clients, each with $b$ samples per batch, is (assuming I.I.D. data) functionally equivalent to training a model with a batch size of $N \times b$.

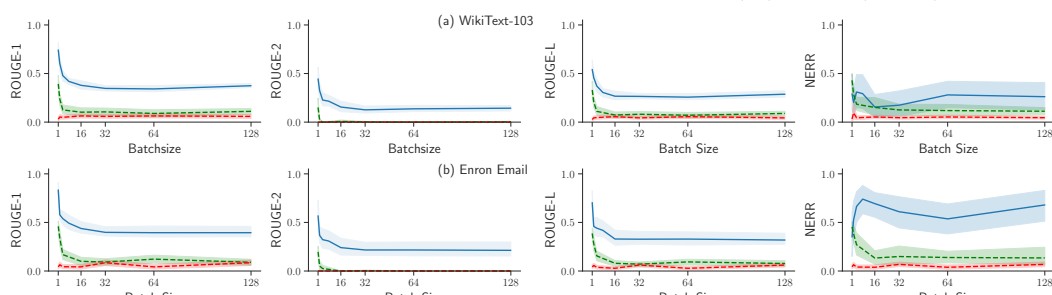

Figure 3: Recovery performance for various batch sizes on WikiText-103 (a) and Enron Email (b). Solid lines indicate the average F-Scores of recoveries out of 20 tested mini-batches. The attack is overall weaker for larger batch sizes (see Table 10 for quantitative results). Moreover, the attack is stronger for the Enron Email dataset.

improves over (Zhu et al., 2019) by prioritizing the matching of gradients of layers that are closer to the input. However, Fig. 3 shows the NERR and ROUGE scores sharply degrade for larger batch sizes. On the contrary, our approach is able to recover a large portion of the original sentences on both datasets even when the batch size is 128. This demonstrates the superior effectiveness of leveraging the knowledge encoded in pre-trained language models and its memorization ability to recover sentences, compared to directly performing optimization in a high-dimensional continuous space.

**Attack performance at different training iterations.** We evaluate the attack performance of our approach on the models trained at different numbers of training iterations $t$, using the WikiText-103 dataset. Specifically, we vary the number of training iterations $t \in \{0, 10000, \ldots, 90000\}$ and fix the batch size for training $b = 16$ and dataset size $N = 200,000$. The batch size for attack is 1.

Previous gradient matching attacks on image data suggests that well-trained models are harder to attack (Geiping et al., 2020) due to the shrinkage of gradient norm as training progresses. However, as shown in Fig. 4 (see qualitative examples in Table 11), we find that well-trained models are more vulnerable to our approach because the model is able to memorize data during training.

**Pre-trained vs randomly-initialized models.** We additionally compare the attack performance for training that starts from a pre-trained model (GPT-2 in our case) and a randomly-initialized model (we use the same GPT-2 architecture but re-initialize all the weights randomly). We find that starting from a pre-trained model is more susceptible to attack than training from scratch as shown in Fig. 4. However, the gap in performance gradually decreases as the number of iterations increases.

As shown in Fig. 4, a pre-trained model which has not been fine-tuned on client data can recover sentences with an average ROUGE-L score of 0.41, indicating that prior knowledge encoded in pre-trained models may help recover sentences. When the model is continued to train on Wikitext-103, the average score increases to 0.53, an almost 30% increase. Our results suggest that leveraging the memorization in the model yields a more powerful attack than only using the language prior encoded in the pre-trained model. We note that we would expect stronger attack performance (especially for smaller training epochs) when attacking larger-capacity models, as stronger models can memorize training instances easily. We report more results for $t = 0$ with different batch sizes in Table 12 in Appendix B.5.

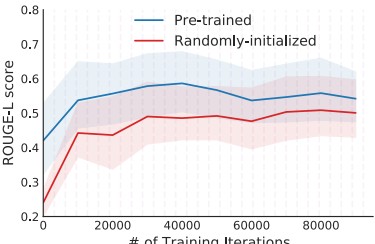

Figure 4: Attack performance over training iterations $t$ for pre-trained or randomly-initialized model on WikiText-103 (batch size for the attack is 1). Dashed lines indicate when an epoch was completed. See Appendix B.5 for qualitative results.

**Recovering multiple sentences by iterative attack.** We further investigate a variant of our attack in which we apply the beam search for multiple repetitions. Each repetition applies an $n$-gram penalty based on the results of previous generations. We consider a sentence to have strong leakage if the longest recovered subsequence is > 25% of the original sentence (i.e., ROUGE-L score > 0.25).

| Iter. | Original Sentence | Recovered Sentence |
|---|---|---|
| 1st | In 1988 it was made into a feature film Katinka, directed by Max von Sydow, starring Tammi Øst as Katinka. | After the end of the year, the film was made into a feature film, and it was directed by Max von Sydow and published by Warner Bros. |
| 2nd | He was killed in a mysterious plane crash on 29 March 1959, while en route to Bangui. | Fox, in an attempt to put an end to the fire, was killed in a plane crash while en route to Guadalcanal, Puerto Rico, on April 6, 2015, while on a |
| 10th | The next year, Edwards published a significant treatise entitled Bibliographic Catalogue of the Described Transformation of North American Lepidoptera. | He had also published a treatise on it, published in 1959, entitled Bibliographic Catalogue of I. |

Table 2: Qualitative results for multi-sentence recovery for a batchsize of 16. The multi-sentence attack is able to recover significant portions of multiple sentences in a single batch.

| (a) Gradient pruning (Zhu et al., 2019) | | | |
| --- | --- | --- | --- |
| Prune ratio | Perplexity | Precision | Recall |
| 0 | 11.46 | 1.00 | 1.00 |
| 0.9 | 11.57 | 1.00 | 1.00 |
| 0.99 | 12.77 | 1.00 | 1.00 |
| 0.999 | 15.34 | 1.00 | 0.98 |
| 0.9999 | 19.21 | 1.00 | 0.90 |

| (b) DPSGD (Abadi et al., 2016) | | | |
| --- | --- | --- | --- |
| $\epsilon$ of DPSGD | Perplexity | Precision | Recall |
| 1 | 16.31 | 0.00 | 0.00 |
| 5 | 14.32 | 0.29 | 0.01 |
| 10 | 12.86 | 0.88 | 0.17 |
| 15 | 11.98 | 0.97 | 0.49 |
| inf. | 11.46 | 1.00 | 1.00 |

Table 4: Precision and recall for the reconstruction of tokens under gradient pruning (a) and DPSGD (b). FILM can still recover a considerable amount of tokens (i.e. >90%) even with a gradient prune ratio of 0.9999. For DPSGD, FILM fails to retrieve the majority of tokens (i.e., recall < 0.5) when $\epsilon$ is smaller than 15.

Table 2 shows a qualitative example of this iterative process. Table 3 shows that for a batch size of 16, an attacker who performs 100 repetitions of beam search is able to recover roughly 34% of the original batch (roughly 5 sentences) with ROUGE-L score over 0.25 (recall). However, since 100 repetitions results in 100 different generated sentences, an attacker would still need a method that picks good recoveries out of them[9]. We estimate the difficulty of choosing good sentences by using the ratio of good sentences to total generated sentences (precision).

| Iter. | Recall | Precision |
| --- | --- | --- |
| 1 | $0.03 \pm 0.03$ | $0.50 \pm 0.51$ |
| 5 | $0.13 \pm 0.08$ | $0.36 \pm 0.22$ |
| 10 | $0.19 \pm 0.11$ | $0.28 \pm 0.16$ |
| 20 | $0.25 \pm 0.12$ | $0.21 \pm 0.11$ |
| 50 | $0.30 \pm 0.11$ | $0.12 \pm 0.04$ |
| 100 | $0.34 \pm 0.09$ | $0.07 \pm 0.02$ |

Table 3: Performance of recovering multiple sentences by applying our attack iteratively on WikiText-103 ($b = 16$). Recall: % of original batch with ROUGE-L>0.25), Precision: % of recovered sentences with ROUGE-L>0.25.

## 6.3 Evaluation of Defenses

We evaluate three defense methods: gradient pruning (Zhu et al., 2019) and Differentially Private Stochastic Gradient Descent (DPSGD) (Abadi et al., 2016), and our method of freezing embeddings. The evaluations use the WikiText-103 dataset and the GPT-2 model, with batch size = 16. Given $S^*$, the set of tokens recovered from the attack and $S$, the original set of tokens in the private batch. Two metrics for performance of the attack are:

1. Precision: $|S^* \cap S|/|S^*|$, which is the fraction of original tokens in the recovered set.
2. Recall: $|S^* \cap S|/|S|$, which is the fraction of recovered original tokens in all original tokens.

We measure the impacts of defenses on model utility by computing the average perplexity of the model across the test set for each dataset. For evaluating the defense of freezing word embeddings during training, precision and recall are both 0.

**Token reconstruction under gradient pruning (Zhu et al., 2019).** Gradient pruning zeros out the fraction of gradient entries with low magnitude. However, the embedding gradients of existing tokens in the batch are still non-zero unless the prune ratio p is extremely high. Thus, the attack strategy remains the same as the vanilla attack: we retrieve the tokens whose embedding gradients are non-zero. As shown in Table 4.a, the attack always returns existing tokens in the private batch (i.e. precision is always 1). The recall decreases as the prune ratio increases, because some entries of word embedding gradients get completely zeroed out and thus the corresponding tokens cannot be retrieved by the attack. However, the attacker can still recover a considerable amount of tokens (i.e. >90%) even with a prune ratio as high as 0.9999.

**Token reconstruction under DPSGD (Abadi et al., 2016).** The strategy to launch the attack under DPSGD is trickier, as the gradients become noisy and the previous heuristics of returning non-zero gradient entries no longer hold. We come up with a new attack strategy which involves using a threshold $\tau = \sigma\sqrt{2 \log d}$ to discriminate noisy gradients with pure noise, where $d$ is the embedding dimension (i.e. 768) and $\sigma$ is the noise scale of DPSGD. For each token, we check the maximum magnitude of its embedding gradients: if the value is larger than $\tau$, then the token may be included in the original batch with high probability. As shown in Table 4.b, the attack performance drops when the epsilon of DPSGD decreases (at the cost of perplexity), because the relative scale between the noise magnitudes and the original gradient magnitude increases.

**Token reconstruction under freezing embeddings.** Freezing embeddings stops the attacker's access to the bag-of-words, and therefore results in *precision and recall both being 0*. However, it is nots considered a standard practice in training language models, as it may lead to worse model utility.

We compare the perplexity of GPT-2 models (1) with frozen embeddings or unfrozen embeddings and (2) from scratch (i.e., from a randomly initialized model) or from pre-trained (i.e., initialized by GPT-2 parameters) in Table 5. We find that freezing word embeddings in training from scratch results in a significant loss of model utility, but is negligible in the setting of using a pre-trained model.

| | From Scratch | | From Pretrained | |
| --- | --- | --- | --- | --- |
| | Unfrozen | Frozen | Unfrozen | Frozen |
| **Wikitext-103** | 27.31 | 118.69 | 11.40 | 11.48 |
| **Enron Email** | 15.16 | 69.17 | 7.09 | 7.30 |

Table 5: Perplexity when embeddings are frozen or unfrozen. We observe a significant drop in perplexity when embeddings are frozen in training from scratch.

# 7 Analysis

We finally discuss the impact of different parameters of our FILM attack in Fig. 5 and Fig. 6. We also study the effect of different training parameters and present the results in Appendix B.7.

**Beam size.** As discussed in Sec. 4.3, beam size (denoted by $k$) is a hyper-parameter of beam search which controls the number of beams active at any time. As shown in Fig. 5 (left), the attack performance grows as the beam size is increased. This result aligns with our intuition using a larger beam size corresponds with consideration of more sentences during search. Considering that the computational resources required for the attack also increases with a larger beam size, we find that a beam size of around 32 achieves the best trade-off in terms of attack performance and efficiency.

**N-gram penalty.** The $n$-gram penalty controls for what size of $n$-gram the penalty is applied. In Fig. 5 (right), we observe that having no penalty performs significantly worse than applying a penalty on repeated 2-grams. We note that performance for $n = 2$ (i.e., penalizing repeated 2-grams) is the closest to the "oracle" case, where the frequency of each word is known from the bag of words.

**Gradient norm's coefficient $\beta$ in prior-based scoring.** As discussed in Sec. 4.4, the prior-based token recording for a sentence $\mathbf{x}$ uses a scoring function $\mathcal{S}(\mathbf{x}) = \text{Perplexity}(\mathbf{x}) + \beta \cdot \text{Gradient Norm}(\mathbf{x})$, where $\beta$ is a hyper-parameter that controls the importance of the gradient norm term. Our results (see Fig. 6 (left)) show that the fine-tuning stage achieves a trade-off between fully perplexity-guided (i.e., $\beta = 0$) and almost fully gradient norm-guided (i.e., $\beta = 50$), with $\beta = 1$ being the optimal. The trend is consistent across different batch sizes.

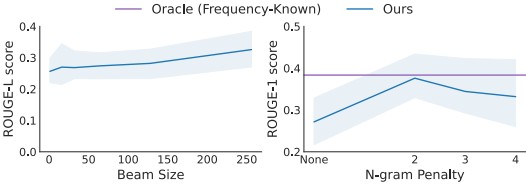
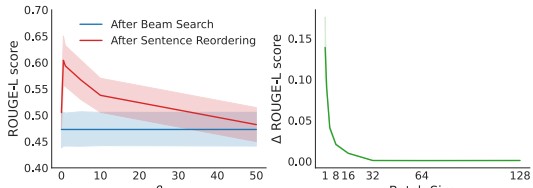

Figure 5: Effect of beam size and $n$-gram penalty. Recovery quality improves with larger beam sizes (left). An $n$-gram penalty ($n = 2$) performs the best (right).

Figure 6: Effect of gradient norm coefficient $\beta$ in the sentence reordering stage (Sec. 4.4) (left), and the improvement of ROUGE-L after fine-tuning (right).

We also notice that the improvement in the token reordering stage is most significant for small batch sizes (see Fig. 6 (right)). We hypothesize this is due to locality; the beam search result for smaller batch sizes is usually closer to its corresponding ground truth sentence than for larger batch sizes, which makes fine-grained reordering more powerful.

# 8 Discussion and Future Work

This paper presents the first attack that can recover multiple sentences from gradients in the federated training of the language modeling task. Our method FILM works well with modern neural language models based on Transformers and large batch sizes (up to 128), raising serious privacy-leakage concerns for federated learning in real-world scenarios. The key insight of our study is that an adversary can gain substantial power to reconstruct sentences by taking advantage of the knowledge memorized by trained language models. We also propose a new defense which freezes the word embeddings during the federated training of language models. Our evaluation shows that it can effectively defend the FILM attack for the setting of starting from a pretrained languages model, with little utility loss.

**Limitations.** The results presented in this paper have several limitations. First, we have not evaluated whether our attack can generalize to other natural language processing tasks, such as text classification. Second, we have not tried our attack for batch sizes beyond 128 and our ablation study does not report results for batch sizes beyond 16. We additionally leave open the question of selecting good sentences in the multi-sentence recovery scenario (Sec. 6.2). Furthermore, the proposed defense of freezing embeddings during the pre-training of the language model can be detrimental to the model utility. Therefore, how to effectively defend the FILM attack for this setting remains open.

**Ethical considerations.** While our intention is to raise the awareness of the privacy risks in the language modeling task in federated learning, from the attacker's prospective, FILM is also likely a step towards even stronger attacks. Therefore, we hope that this work can also motivate a necessary redesign of defenses to provide meaningful privacy guarantees to clients for training language models in federated learning.

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
