# A Beam Search Algorithm

Algorithm 1 demonstrates the step-by-step operations of our beam search algorithm (see Sec. 4.3).

---

**Algorithm 1:** Beam search to order words

---

**Input:** Bag of words $B = \{w_1, \ldots, w_m\}$,
A maximum sentence length $L$, beam size $k$,
A set of $z$ possible prompts $P = \{p_1, \ldots, p_z\}$,
Distribution defined by the trained language model $\mathbb{P}_{\theta^t}$,
$n$-gram count function $R_n$ with associated penalty $\rho$,
**Output :** $g * k$ sentences consisting of words in $B$

1 $\pi^1 \leftarrow P$ **for** $i \in [L]$ **do**
2 $\quad$ $\pi^{i+1} \leftarrow \emptyset$
3 $\quad$ **foreach** $(beam, word)$ *in* $\pi^i \times B$ **do**
4 $\quad\quad$ $\pi^{i+1} \leftarrow \pi^{i+1} \cup \{\text{CONCAT}(beam, word)\}$
5 $\quad$ **end**
6 $\quad$ $\pi^{i+1} \leftarrow \text{TOP}(\pi^{i+1}, k, f_{\theta^t})$
7 **end**
8 **return** $\pi^L$;

1 **Procedure** $\text{Top}(\pi, k, \mathbb{P}_{\theta^t})$:
2 $\quad$ SCORES $\leftarrow \emptyset$
3 $\quad$ SCORES $\leftarrow$ SCORES $\cup \log \mathbb{P}_{\theta^t}(beam) - \rho R_n(beam)$
4 $\quad$ **return** *Top $k$ elements of $\pi$ ordered by* SCORES

---

# B Experimental Details and More Results

## B.1 Preprocessing details for WikiText-103 and Enron Email

We consider recovering sentences in the current work. As WikiText-103 and Enron Email have longer context (i.e. paragraph), in our evaluation, we split them into sentences and only keep the sentences with 1) fewer than 40 tokens, 2) fewer than 5 unknown tokens, where unknown tokens are tokens not in the vocabulary of the GPT-2 model, 3) a perplexity smaller than 50 on the pretrained GPT-2 model, 4) at least 1 named entities[10] of type: "PERSON", "ORG", "GPE", "LOC", "PRODUCT", "EVENT". We leave recovering longer paragraphs as future work.

We keep 2000 examples of each dataset as the evaluation set, and use the left for training.

## B.2 Comparison with Previous Attacks

Table 6 presents a high-level comparison of key differences between our FILM attack and previous attacks. Our FILM attack is unique as it does not rely on end-to-end optimization, is demonstrated on large batch sizes, and is focused on the autoregressive language modelling task.

| Name | Technique | max($\mathbf{b}$) | Model(s) | Datasets (Sequence Length, Task) |
|---|---|---|---|---|
| DLG (Zhu et al., 2019) | E2E | 1 | BERT | Masked Language Modeling ($\sim 30$) |
| TAG (Deng et al., 2021) | E2E + Reg | 1 | TinyBERT, BERT, BERTLARGE | CoLA (5-15, Sentence Classification), SST-2 (10-30, Sentiment Analysis), RTE (50-100, Textual Entailment) |
| LAMP (Dimitrov et al., 2022) | E2E + Reg + DR | 4 | TinyBERT, BERT, BERTLARGE | CoLA (5-9) SST-2 (3-13), Rotten Tomatoes (14-27, Sentiment Analysis) |
| FILM (Ours) | BoW + BS + DR | 128 | GPT-2 | Wikitext-103, Enron Email (15-40, Autoregressive Language Modeling) |

Table 6: A high-level overview of the key differences between FILM and prior work on extracting information from gradients in federated language modelling. $\max(\mathbf{b})$ is the maximum attack-able batch size. E2E means "End-to-End optimization", "Reg" means the inclusion of a regularization term, "DR" refers to a discrete token reordering step, and "BoW"refers to bag-of-words reordering. Our approach is unique as it does not rely on end-to-end optimization, is demonstrated on large batch sizes (i.e. larger $\max(\mathbf{b})$), and is focused on the autoregressive language modelling task.

---

[10]We identify named entities by SpaCy (Honnibal et al., 2020).

## B.3 More Results of DLG (Zhu et al. 2019) and TAG (Deng et al. 2021)

Table 7 shows the attack results of Zhu et al. (2019) with different batch sizes. Their gradient-matching optimization fails to recover sentences with different batch sizes. We also find that Zhu et al. (2019) seem sensitive to the initialization. Table 8 shows the attacking results with different initialization. We find that only when the initialization is very close to the original sentence (i.e., one- or two-word different), the attack can recover sentences successfully. Table 9 compares metrics for reconstructions by previous methods and by ours

| Batch Size $b$ | Original sentence | Best Reconstructed sentence |
|---|---|---|
| $b = 1$ | As Elizabeth Hoiem explains, "The most English of all Englishmen, then, is both king and slave, in many ways indistinguishable from Stephen Black. | thelesshovahrued theAsWords reporting the Youngerselagebalance, mathemat mathemat mathemat reper arrangpmwikiIndia Bowen perspectoulos subur, maximal |
| $b = 2$ | As Elizabeth Hoiem explains, "The most English of all Englishmen, then, is both king and slave, in many ways indistinguishable from Stephen Black. | addons, Rosesineries,-, princ,soDeliveryDate Aires gazed,.ropolitan glim eventscffff Americans hereditary vanishing traged defic mathematenegger levied mosquodan: antioxid mathematetheless Wh |
| $b = 4$ | The expressway progresses northward from the onramp, crossing over Waverly Avenue and passing the first guide sign for exit 2(NY 27), about 0@. | Thebridsptoms Rainbow. plotted the Gleaming,. scrutlocked and.. apex llularetheless Emailoen the explan challeng. treatedFormer pieces government |
| $b = 8$ | Hagen believes that despite the signifying that occurs in many of Angelou's poems, the themes and topics are universal enough that all readers would understand and appreciate them. | of,arning plaint sacrific Protestant the..Medical littleisha.isky,wallve ointed way skeletodor aestorydemocratic. enclaveiHUDThe repetitionTrivia useful |

Table 7: Reconstruction performance of Zhu et al. (2019)'s attack with different batch sizes. For each batch size, we show the best-case reconstructed sentence across 10 evaluated batches.

| Initialization | Reconstructed sentence |
|---|---|
| He at ordered the construction of Fort Oswego at the mouth of the Oswego River. | He consequently ordered the construction of Fort Oswego at the mouth of the Oswego River. |
| Hefly ordered the construction of Fort Os contributionsgo at the halls Oath the Oswego River. | Hehower consequently the construction ofinterstitial ranchyr the mathemat Adinemort Fort Os annot mathema |
| itaire smash Full Mongo Highly C sphere Commodore intermediate report subjug WROf Anti Samueldet backward sec Kill manufacturer | thelesstheless mathematMod Loaderastedperty perpendkefellerDragonMagazine horizont mathematoperative pediatricsoDeliveryDatesoDeliveryDate mathemat mathemat manufacturer |

Table 8: Reconstruction performance of Zhu et al. (2019)'s attack with different initialization. The original sentence is "He consequently ordered the construction of Fort Oswego at the mouth of the Oswego River.". Text in green represents the words that are recovered successfully.

| | | B=1 | | | | B=4 | | | | B=16 | | | | B=32 | | | |
|---|---|---|---|---|---|---|---|---|---|---|---|---|---|---|---|---|---|
| | | R-1 | R-2 | R-L | NERR | R-1 | R-2 | R-L | NERR | R-1 | R-2 | R-L | NERR | R-1 | R-2 | R-L | NERR |
| **WikiText-103** | DLG | 0.03 | 0.00 | 0.03 | 0.03 | 0.05 | 0.00 | 0.05 | 0.04 | 0.06 | 0.00 | 0.06 | 0.05 | 0.06 | 0.00 | 0.04 | 0.05 |
| | TAG | 0.39 | 0.15 | 0.33 | **0.43** | 0.13 | 0.00 | 0.11 | 0.18 | 0.10 | 0.01 | 0.07 | 0.15 | 0.11 | 0.00 | 0.08 | 0.13 |
| | FILM | **0.74** | **0.44** | **0.54** | 0.25 | **0.48** | **0.23** | **0.37** | **0.26** | **0.38** | **0.16** | **0.27** | **0.25** | **0.35** | **0.13** | **0.26** | **0.24** |
| **Enron Email** | DLG | 0.04 | 0.00 | 0.04 | 0.04 | 0.04 | 0.00 | 0.04 | 0.04 | 0.04 | 0.00 | 0.03 | 0.04 | 0.09 | 0.00 | 0.06 | 0.07 |
| | TAG | 0.46 | 0.20 | 0.39 | **0.45** | 0.17 | 0.03 | 0.16 | 0.27 | 0.10 | 0.00 | 0.08 | 0.13 | 0.09 | 0.00 | 0.07 | 0.15 |
| | FILM | **0.83** | **0.57** | **0.70** | 0.35 | **0.54** | **0.32** | **0.44** | **0.66** | **0.44** | **0.24** | **0.33** | **0.69** | **0.40** | **0.22** | **0.33** | **0.61** |

Table 9: A comparison of text reconstructions from gradients for various datasets, prior methods, and batch sizes (denoted by B). R-1, R-2, and R-L, denote average ROUGE-1, ROUGE-2 and ROUGE-L scores, respectively. All data represents average values collected from 20 samples. Our method (FILM) is able to recover more data from sentences across all batchsizes and datasets than prior methods. We additionally note the significant gap in ROUGE-2 scores between FILM and prior methods, indicating significantly better recovery of ordering of words in sentences.

## B.4 Quantitative Results for Different Batch Sizes

Table 10 shows all four metrics for the attack results with different batch sizes; Fig. 7 shows the performance of the best reconstructions for each metric and batchsize.

| Batch Size | ROUGE-1 | ROUGE-2 | ROUGE-L | NERR |
|---|---|---|---|---|
| **WikiText-103** | | | | |
| 1 | $0.74 \pm 0.12$ (1.00) | $0.44 \pm 0.19$ (1.00) | $0.54 \pm 0.14$ (1.00) | $0.28 \pm 0.35$ (1.00) |
| 16 | $0.38 \pm 0.08$ (0.61) | $0.16 \pm 0.10$ (0.44) | $0.27 \pm 0.08$ (0.47) | $0.26 \pm 0.25$ (1.00) |
| 64 | $0.34 \pm 0.06$ (0.52) | $0.14 \pm 0.06$ (0.30) | $0.26 \pm 0.05$ (0.40) | $0.36 \pm 0.19$ (0.67) |
| 128 | $0.37 \pm 0.04$ (0.50) | $0.14 \pm 0.05$ (0.24) | $0.29 \pm 0.06$ (0.50) | $0.28 \pm 0.19$ (0.50) |
| **Enron Email** | | | | |
| 1 | $0.73 \pm 0.21$ (1.00) | $0.45 \pm 0.29$ (1.00) | $0.61 \pm 0.24$ (1.00) | $0.34 \pm 0.45$ (1.00) |
| 16 | $0.41 \pm 0.09$ (0.81) | $0.20 \pm 0.12$ (0.74) | $0.31 \pm 0.10$ (0.81) | $0.40 \pm 0.31$ (1.00) |
| 64 | $0.37 \pm 0.08$ (0.74) | $0.18 \pm 0.10$ (0.70) | $0.29 \pm 0.10$ (0.74) | $0.35 \pm 0.26$ (1.00) |
| 128 | $0.37 \pm 0.07$ (0.72) | $0.19 \pm 0.10$ (0.68) | $0.29 \pm 0.09$ (0.72) | $0.38 \pm 0.31$ (1.00) |

Table 10: Reconstruction performance for various batch sizes on WikiText-103 and the Enron Email dataset. The table reports the average metric values of reconstructions out of 20 tested mini-batches with the standard deviation; the best results are shown in parenthesis.

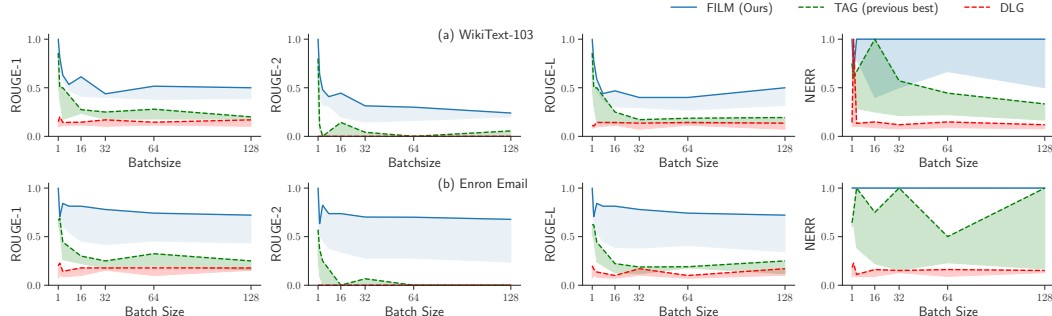

Figure 7: Recovery performance of the best sentences for various batch sizes on WikiText-103 (a) and the Enron Email dataset (b). Solid lines indicate the average F-Scores of recoveries out of 20 tested mini-batches.

## B.5 More Results For Attacking the Pre-trained Model

Table 11 presents attack results at different training iterations of a pretrained model. Similar to our finidngs in Sec. 6, FILM recovers better sentences when $t$ increases.

Table 12 provides more examples for running our attack on a pre-trained language model (i.e. without any training with private data) with different batch sizes. The attack is relatively weak in this scenario, showing that leveraging the memorization in the model give a much more powerful attack than only using the language prior encoded in the pre-trained model.

| | |
|---|---|
| **Original**: The short@-@tail stingray forages for food both at night and during the day. | |
| $t$ | **The recovered sentence** |
| 0 | The, short for short-tail stingray at night during night day during day at day night at |
| 10,000 | The stingray forages for food at night both at both during food food shortages at short food |
| 20,000 | The short@-@tail stingray forages for food during the day and for the night. |
| 40,000 | The short@-@tail stingray forages for food both at night and during the day. |

Table 11: An illustration of the recovered sentence with different training iterations $t$ (batch size = 1) with a pretrained model. Text in green represents the phrases and words that are recovered successfully. Our attack recovers better sentences when $t$ increases.

| Batch Size $b$ | Original sentence | Best Reconstructed sentence |
|---|---|---|
| $b = 1$ | Though Chance only batted.154 in the 1907 World Series, the Cubs defeated the Tigers in four games. | Though in the World Series, the Cubs defeated the Tigers in four games. |
| $b = 2$ | I thought if I did the animation well, it would be worth it, but you know what? | I, you know, I know what I did, but I thought, well, what would it be if you did it? |
| $b = 4$ | In 1988 it was made into a feature film Katinka, directed by Max von Sydow, starring Tammi Øst as Katinka. | Since it was made into a film starring Max von Sydow as Tammi von T. |
| $b = 8$ | I thought if I did the animation well, it would be worth it, but you know what? | By the time I did it, I thought, well, you know, what's the point of it? |

Table 12: Reconstruction performance of our attack when $t = 0$ with different batch sizes. For each batch size, we show the best-case reconstructed sentence across 10 evaluated batches measured by the ROUGE score. Text in green represents the words that are recovered successfully.

## B.6 Qualitative Results with Randomly Initialized Models

Table 13 presents attack results at different training iterations of a randomly initialized model. The overall attack quality is lower than the results with a pre-trained model.

| **Original**: The short@-@tail stingray forages for food both at night and during the day. | |
|---|---|
| $t$ | **The recovered sentence** |
| 0 | The' the the at at night night. |
| 10, 000 | The short@-@ray for the day and at the night for both the and the for@ |
| 20, 000 | The at@-@ray for the night at the and at night the day and the for@ |
| 40, 000 | The@-@ray for the day during both. |
| 80, 000 | The@-@tail stingray forages for food for both for short foodages both both food. |

Table 13: An illustration of the recovered sentence with different training iterations $t$ (batch size = 1) with a randomly initialized model. Text in green represents the phrases and words that are recovered successfully. Our attack recovers better sentences when $t$ increases. However, the overall attack quality is lower than the results with a pre-trained model (see Table 11).

## B.7 Ablation Study for Training Configurations

We also perform an ablation study on WikiText-103 to investigate how the attack scales with different training parameters, including the size of the training dataset and the initial learning rate.

**Larger training set sizes are not harder to attack.** It was believed that training with large training sets helps avoid over-fitting and generalizes better as it captures the inherent data distribution more effectively, while training with small datasets may result in the model memorizing the training set (Arpit et al., 2017)—which in our case, may yield a better attack performance. However, our experiments suggest that larger training set sizes are *not* any more difficult to attack than smaller training sets. As shown in Fig. 8.a, the attack performance is quite similar across different evaluated training set sizes, from $5, 000$ examples to $200, 000$.

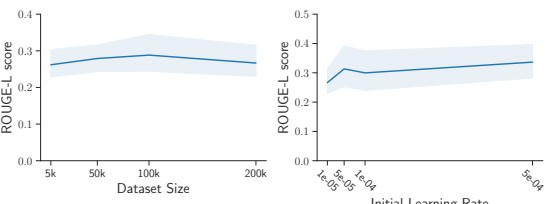

Figure 8: Performance of our attack under different training configurations for the WikiText-103 dataset, including the size of the training dataset (a) and the initial learning rate (b). We note that the recovery quality slightly improves with larger initial learning rates. Surprisingly, the recovery quality is flat across different dataset sizes.

**Larger initial learning rate is more vulnerable to attack.** We also show how the attack performs with with different initial learning rates by varying the initial learning in $\{1 \times 10^{-5}, 5 \times 10^{-5}, 1 \times 10^{-4}, 5 \times 10^{-4}\}$ while fixing the batch size $b = 16$ and dataset size $N = 200, 000$. As we apply early stopping during training, none of the models we test were overfit. As shown in Fig. 8.b, as the initial learning rate increases, our attack is able to achieve slightly higher ROUGE scores.

Table 14 presents the attack results recovered under different initial learning rates. We find that as the initial learning rate increases, our attack is able to successfully reconstruct more components of the training sentence despite similar final training and testing perplexity.

| Initial LR | Train/Test Perplexity | Original Sentence | Reconstructed Sentence |
|---|---|---|---|
| $1e-5$ | 7.60/11.11 | A tropical wave organized into a distinct area of disturbed weather just south of the Mexican port of Manzanillo, Colima, on August 22 and gradually moved to the northwest. | Early on September 22, an area of disturbed weather organized into a tropical wave, which moved to the northwest of the area, and then moved into the north and south@-@to the northeast. |
| $5e-5$ | 6.05/11.56 | A tropical wave organized into a distinct area of disturbed weather just south of the Mexican port of Manzanillo, Colima, on August 22 and gradually moved to the northwest. | Early on September 22, a tropical wave moved into the area, which organized into an area of disturbed weather just south of the Mexican port of Manzanillo, and moved to the south@-@ |
| $1e-4$ | 6.03/12.21 | Kristina Lennox@-@Silva also represented Puerto Rico as a female swimmer in the 400 meters freestyle. | Kristina Lennox@-@Silva also represented Puerto Rico as a female swimmer in the 400 meters freestyle at the 2015 Miss New Japan Pro Wrestling Tag Team Championship, and represented Japan at |
| $5e-4$ | 6.87/10.95 | I thought if I did the animation well, it would be worth it, but you know what? | I thought it would be worth it if I said I did it to you and you know what you did. |

Table 14: An illustration of the reconstructed sentence with different initial learning rate (batch size $b = 16$); All models are trained using early stopping. Text in green represents the phrases and words that are recovered successfully. Our attack seems to give better performance with a larger initial learning rate.

## B.8 Justification for the Scoring Function in the Reordering Stage

As discussed in Sec. 4.4, our scoring function consists of a gradient norm term and a perplexity term (see Eq.4). This design is mainly motivated by the observation that sentences in the training corpus usually have lower perplexity scores and lower gradient norms on the trained language model, compared with their slightly altered versions.

To demonstrate this, we randomly pick the 5 following sentences from the training corpus:

- *"Significant increases in sales worldwide were reported by Billboard in the month of his death.";*
- *"The office has subsequently been held by one of the knights, though not necessarily the most senior.";*
- *"In 1999, several Russian sources reported that Laika had died when the cabin overheated on the fourth orbit.";*
- *"Officials advised 95@,@000 citizens along the New Jersey coastline, an area that rarely experiences hurricanes, to evacuate.";*
- *"It later won a 2014 Apple Design Award and was named Apple's best iPhone game of 2014.";*

For each original sentence, we generate its slightly altered versions which performs *one* of the following three operations (we generate 10 sentences for each operation):

- **Swap** two randomly selected tokens in the original sentence;
- **Delete** a randomly selected token from the original sentence;
- **Insert** a token (randomly selected from the bag of tokens of the original sentence) into a random position of the original sentence.

As shown in Fig. 9, original sentences usually have lower gradient norm and perplexity score than their corresponding slightly altered versions.

## B.9 How much does the bag of words help?

We briefly compare the performance of the attack in a scenario where beam search is performed with the bag of words (as in FILM), and without the bag of words (i.e. if an attacker only had the model, but not bag of words). Both settings were ran with the same prompts for all samples.

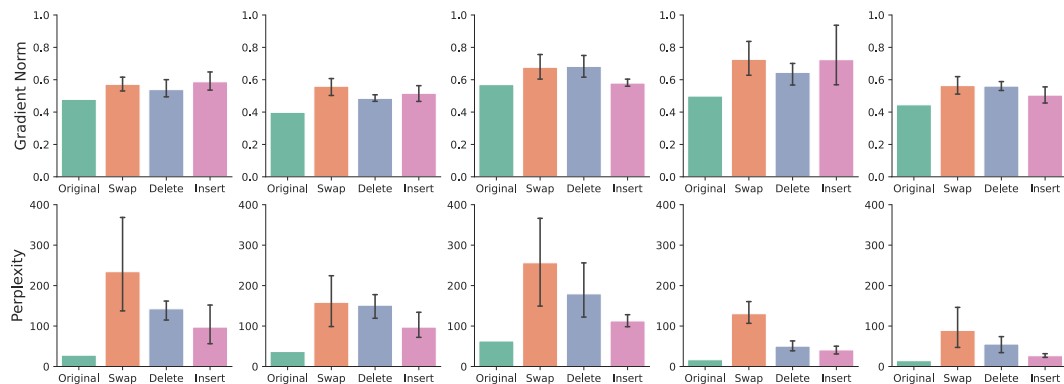

Figure 9: Gradient norm (the first row) and perplexity (the second row) for 5 original sentences and their corresponding slightly altered sentences (by swapping, deletion or insertion) on the trained language model. Original sentences usually have lower gradient norm and perplexity.

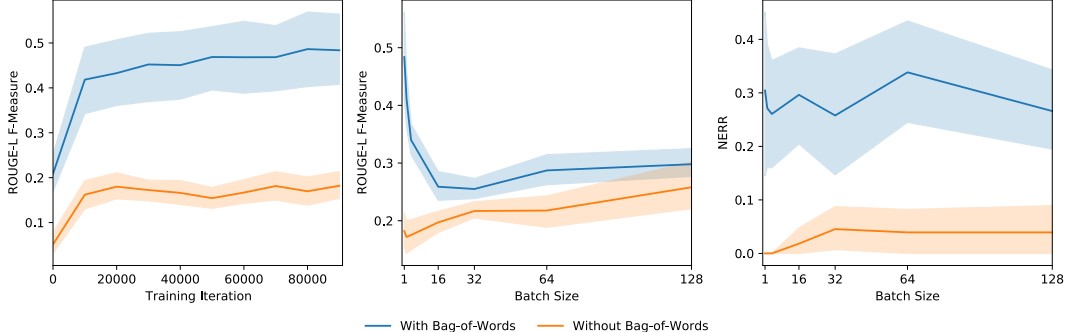

Figure 10: Performance of using beam search with and without a bag of words to recover memorized sentences from GPT-2 finetuned on WikiText-103. We compare over different training iterations with a batchsize of 1 (left), and across different batch sizes at 90,000 iterations (middle). We also include the NERR score at 90,000 iterations (right). Overall we find that beam search is significantly stronger for small batchsizes if the attacker also has a bag of words. Moreover, using the bag of words allows attackers to reconstruct significantly more named entities even in larger batch sizes.