# OpenReview forum: "Recovering Private Text in Federated Learning of Language Models"
_NeurIPS.cc/2022/Conference — NeurIPS 2022 Accept_

### Official Review · Reviewer_vs2F · 2022-07-07

**Rating:** 4
**Confidence:** 5
**Soundness:** 3 good
**Presentation:** 3 good
**Contribution:** 2 fair

**Summary:**

This paper proposes a new attack named FILM for extracting text from federated learning models.
The key idea is to use gradients with respect to embedding matrix in order to identify words occurring in the batch,
and then use beam search and token reordering to reconstruct a sentence from the batch.
The method is experimentally evaluated on several datasets which show benefits compared to the baseline.


**Questions:**

1) The authors say they focus on extracting 1 sentence from the batch, but could they evaluate how well can they recover full batches, as that is the setting in other papers?

2) It would be good to see some analysis of which sentence in the batch is actually recovered. Is it always the shortest sentence?

3) As mentioned earlier, this attack is easily bypassed by a defense which does not train word embeddings which prevents reconstructing bag of words. Could you experiment with this defense and some other defenses (e.g. differential privacy based)?


**Limitations:**

Yes.

**Strengths And Weaknesses:**

I believe that this paper considers quite relevant problem as many practical applications of federated learning are in NLP space, while
existing attacks have mostly focused on reconstructing images.

However, I have found some issues with the paper which I hope authors can clarify. Due to these issues I currently think the paper is below the acceptance bar.

First, there is a serious limitation to this approach which I think should be emphasized.
This is the fact that the extraction approach described in Section 4.2 assumes that token embeddings are also being trained.
When embeddings are not trained, this approach doesn't work at all as it cannot compute bag of words in the batch.
Furthermore, this limitation does not exist in other papers that consider this setting, e.g. DLG (Zhu et al.), TAG (Deng et al.), LAMP (Dimitrov et al.) as they can utilize other elements of the gradient.
Therefore, since this attack is bypassed by just not training embeddings (which is sometimes even done by default), it is unclear how significant it is.

First, comparison with TAG (Deng et al.) and DLG (Zhu et al.) does not seem to match results in respective papers. Results in Table 1 show that DLG and TAG essentially can't reconstruct anything even for batch size 1.
Moreover, authors claim that TAG doesn't improve over DLG (footnote in Table 1). This all clashes with results in Deng et al. and Dimitrov et al. where DLG has much better reconstruction than shown in this submission,
and using TAG further improves the reconstructions.
In this paper, authors evaluate on completely new datasets than the prior papers (Zhu et al., Deng et al., Dimitrov et al.) so it is difficult to compare to those works, and might be a reason for this discrepancy.
I would suggest authors to experiment with datasets used in other papers (e.g. CoLA, SST-2, Rotten Tomatoes) so that we can compare the actual performance of different methods.

In Section 4.4 I find it a bit strange to optimize for the sentence which minimizes combination of prior and gradient norm.
Why should we not combine prior with L2 distance between the current gradient and the ground truth gradient, as in Zhu et al.?
Could authors experimentally demonstrate that their loss function works better?

Furthermore, Section 4.4 seems basically the same approach as discrete optimization from Dimitrov et al.
The only difference I can see is that this paper is using gradient norm instead of the L2 distance between the gradients (which is not clear why this works better, as discussed above).
Given that authors are aware of that work, they should acknowledge in Section 4.4 that this approach comes from Dimitrov et al., and discuss the differences.

As a more general comment, the paper at few places does not acknowledge contributions from other paper and tends to overclaim their own contributions:

- As mentioned earlier, Section 4.4 is basically the same as in LAMP (Dimitrov et al.), and yet there is no acknowledgment of this.
- In the paragraph starting on Line 244 authors write that they propose using ROUGE as a metric, but this was already used in TAG (Deng et al.) and should be acknowledged there.
- In L327, authors write "This paper presents the first attack that can recover sentences from gradients in federated learning of language models", but this is not true as such reconstruction has already been performed by Zhu et al., Deng et al., Dimitrov et al.

According to this, writing should change to account from contribution from other papers.

Typo:

L71: both work -> both works

---

> ### Author Response · Authors · 2022-08-02
> **Responses to Reviewer vs2F (1/2)**
>
> We thank the reviewer for the helpful feedback. We have tried to address all your comments in the following. Please let us know if you feel we haven’t fully addressed your comments. We will be happy to address them further.
>
> # Q1. Freezing embedding layers as a defense & other defenses (e.g. differential privacy based)?
>
> Please refer to [response to common question #2](https://openreview.net/forum?id=dqgzfhHd2-&noteId=necwm689O6D) for the utility drop induced by freezing embedding, and [response to common question #4](https://openreview.net/forum?id=dqgzfhHd2-&noteId=PU9MOie6ywKn) for results for defenses such as DPSGD and gradient pruning.
>
> # Q2. Comparison with TAG (and DLG)?
> Please refer to [response to common question #1](https://openreview.net/forum?id=dqgzfhHd2-&noteId=smNM_fTegn).
>
> # Q3. Questions about our Section 4.4
>
> > Section 4.4 seems basically the same approach as discrete optimization from Dimitrov et al. The only difference I can see is that this paper is using gradient norm instead of the L2 distance between the gradients (which is not clear why this works better, as discussed above).
>
> Thanks for sharing the concern. However, we respectfully disagree with the reviewer, and would like to highlight that our reordering methods in Section 4.4 are **fundamentally different** from Dimitrov et al.’s approach:
> * The reordering step of our Section 4.4 takes as input a **single sentence** (recovered from beam search in Section 4.3); the goal of this step is to improve this sentence’s quality. Since the attacker only has access to gradients of full batch (instead of gradients of individual sentences), we propose to use some priors to guide the reordering, including 1) the single-sentence gradient norm, and 2) the sentence perplexity.
> * However, Dimitrov et al.’s approach takes as input the batched gradient; their goal is to  run an end-to-end optimization to recover the **whole batch of sentences**. Therefore they can use the L2 distance between the gradients as the optimization goal.
>
> The motivation for using beam search (our major technical contribution) + single-sentence reordering rather than batch-wise end-to-end optimization is: we found that the gradient-matching based approaches (Zhu et al., Deng et al.) don’t work well in recovering data for LMs as they are highly sensitive to initializations (see Appendix C.1). Appendix C.6 provides empirical analysis that supports the design of our reordering method. Moreover, the usage of only the embedding gradients and the language model represents a new attack vector which should be considered separate from that of end-to-end optimization.
>
> We will update our writing to make this distinction clearer.
>
> > In Section 4.4 I find it a bit strange to optimize for the sentence which minimizes combination of prior and gradient norm. Why should we not combine prior with L2 distance between the current gradient and the ground truth gradient, as in Zhu et al.? Could authors experimentally demonstrate that their loss function works better?
>
> This is because that the attacker doesn’t have access to the ground truth gradient for a single sentence in the batch (only the aggregated gradients), and thus we are not able to obtain the per-sentence L2 distance between the current gradient and the ground truth gradient.
>
> # Q4. Justification for our contribution
> > Authors write "This paper presents the first attack that can recover sentences from gradients in federated learning of language models", but this is not true as such reconstruction has already been performed by Zhu et al., Deng et al., Dimitrov et al.
>
> As discussed in Table 2 (see [response to common questions #1](https://openreview.net/forum?id=dqgzfhHd2-&noteId=smNM_fTegn)), the attack by Zhu et al. and Deng et al. can only recover a single sentence (an results in Table 1 in our manuscript show that DLG and TAG usually work poorly even for batch size 1); the attack by Dimitrov et al. can recover up to 4 sentences but is designed for the text classification task. Thus, our attack is the first attack that demonstrates successful recovery of **sentences** (rather than a single sentence) from gradients in federated learning of **autoregressive** language models (rather than text classification models).
>
> To avoid confusion, we’ve updated the sentence to be “This paper presents the first attack that can recover **multiple sentences** from gradients **in the federated training of the language modeling task**” (see lines 333 - 334).
>
> > In the paragraph starting on Line 244 authors write that they propose using ROUGE as a metric, but this was already used in TAG (Deng et al.) and should be acknowledged there.
>
> Thanks for pointing this out; we didn’t mean to claim ROUGE as a new metric for evaluating the attack. We’ve updated the manuscript to avoid this confusion (see lines 250 - 254).

---

> > ### Author Response · Authors · 2022-08-02
> > **Responses to Reviewer vs2F Cotd. (2/2)**
> >
> > # Q5. Performance for multiple sentences
> > > The authors say they focus on extracting 1 sentence from the batch, but could they evaluate how well can they recover full batches, as that is the setting in other papers?
> >
> > Thanks for the question. Our attack recovers only a single sentence at a time. We evaluate our method on recovery of multiple sentences from the same batch in Table 3 in our manuscript. While our method does not reliably reconstruct every single sentence in a batch, the recall metric in Table 3 demonstrates the fraction of a batch we are able to reconstruct to a satisfactory degree. In general, we do not consider recovery of a full batch to be as important of a milestone; Recovery of even a single sentence from a batch should be considered severe leakage by itself.
> >
> > # Q6. Analysis of Length of Recovered Sentences
> > > It would be good to see some analysis of which sentence in the batch is actually recovered. Is it always the shortest sentence?
> >
> > Thanks for the insightful question. We find that, for a batch of size 128, the original sentence which best matches the reconstruction is on average 29 tokens long (std deviation 8.5). This indicates that memorization occurs even for longer sentences. Previous work [1] indicates that memorization is actually stronger in scenarios where the model is prompted with a longer context.
> >
> > # Minor comments:
> > - Fixed the typo in line 71
> >
> > # References
> > [1] N. Carlini, D. Ippolito, M. Jagielski, K. Lee, F. Tramer, and C. Zhang, “Quantifying Memorization Across Neural Language Models.” arXiv, Feb. 24, 2022. doi: 10.48550/arXiv.2202.07646.

---

> > > ### Comment · Reviewer_vs2F · 2022-08-08
> > > **Response**
> > >
> > > Thanks for the response. I think many of my questions are addressed, but I am still not quite convinced about the defense which does not train token embeddings. The authors provided some numbers, but I think it would be necessary to back this with some citations that confirm such drastic loss in performance when not training embeddings, as well as more thorough experiments with this setup.

---

> > > > ### Author Response · Authors · 2022-08-08
> > > > **Response to follow-up**
> > > >
> > > > Thanks for the response. We are glad to learn that we've addressed most of your questions!
> > > >
> > > > Regarding the follow-up question, we completely agree with the reviewer that it is worthwhile discussing freezing embedding as a defense; we plan to incorporate it into our final version.
> > > >
> > > > > I think it would be necessary to back this with some citations that confirm such drastic loss in performance when not training embeddings, as well as more thorough experiments with this setup.
> > > >
> > > > **a. Supporting references**. Training token embeddings has been a standard practice in training large language models [1, 2, 3] from scratch. It has also been shown that continuing training LM with embedding updates leads to large performance gains, especially for low-resource settings and for adapting to different domains [4].
> > > >
> > > > **b. More experiments with this setup**.  We found the final perplexity on the test set to be significantly worse (68.37 with frozen embeddings vs 15.96 with unfrozen embeddings) on the Enron email dataset (one of the two datasets we used to evaluate our attack). Given that the rebuttal window is closing very soon (in less than a day), we are more than happy to provide more results with other language modeling datasets in the final version.
> > > >
> > > > Please let us know if there are any additional questions we can address!
> > > >
> > > >
> > > >
> > > >
> > > > **References:**
> > > >
> > > > [1] Radford, A., Wu, J., Child, R., Luan, D., Amodei, D., Sutskever, I., et al. Language models are unsupervised multitask learners. OpenAI blog 2019.
> > > >
> > > > [2] Brown, T., Mann, B., Ryder, N., Subbiah, M., Kaplan, J.D., Dhariwal, P., Neelakantan, A., Shyam, P., Sastry, G., Askell, A. and Agarwal, S.. Language models are few-shot learners. NeurIPS 2020
> > > >
> > > > [3] Chowdhery, A., Narang, S., Devlin, J., Bosma, M., Mishra, G., Roberts, A., Barham, P., Chung, H.W., Sutton, C., Gehrmann, S. and Schuh, P., 2022. Palm: Scaling language modeling with pathways. arXiv preprint arXiv:2204.02311.
> > > >
> > > > [4] Gururangan, S., Marasović, A., Swayamdipta, S., Lo, K., Beltagy, I., Downey, D. and Smith, N.A., 2020. Don't stop pretraining: adapt language models to domains and tasks. arXiv preprint arXiv:2004.10964.

---

> ### Author Response · Authors · 2022-08-08
> **Thanks again and following up**
>
> Dear Reviewer vs2F,
>
> Thank you again for your thoughtful review. We are very thankful for your comments and suggestions that helped improve our paper and have reflected them in our revision.
>
> As the end of the discussion is approaching, it will be great if we can learn whether our responses have addressed your concerns. We would also appreciate the opportunity to engage further if needed.
>
> Kind regards,
>
> Authors of Paper10606

---

### Official Review · Reviewer_FFKN · 2022-07-10

**Rating:** 5
**Confidence:** 4
**Soundness:** 2 fair
**Presentation:** 3 good
**Contribution:** 2 fair

**Summary:**

This paper proposed FILM, an inversion attack that reconstructs text given gradients and language model parameters in the federated learning setting. The threat model considered an adversarial eavesdropper who can extract gradients and model parameters. FILM first recovers the bag-of-words from gradients and then exploits the model memorization to reconstruct the order of the text. The attack was evaluated empirically on benchmark datasets and outperformed prior attacks that were proposed to recover images. No defense was evaluated against the proposed attack.


**Questions:**

1. Another common threat model is to have a malicious server who knows only about the gradients aggregated on multiple users on multiple batches. How would FILM work in such a threat model?
2. I would appreciate it if authors could provide an intuition behind using gradient norm in the scoring function in Section 4.4 instead of appendix so that readers could understand the method more easily.


**Limitations:**

The authors acknowledged their limitations but I think the paper can be improved if there were quantitative measurements on the limitations.


**Strengths And Weaknesses:**

Strengths:
1. The paper is well-organized. The threat model and the attack methods are explained clearly and easy to follow.
2. The attack method is simple yet exploits the key vulnerabilities in training language models (i.e. memorization so that model would rank training data higher). The comparison with prior work also demonstrated that the attack is effective.

Weaknesses:
1. Although the author acknowledged this as well, there was no evaluation on any mitigation. I think it is critical to know the privacy-utility tradeoff for different mitigation strategies to understand the real impact of any attack.
2. Seems like there are also other easy defenses which the author did not discuss. 1). If the server and client encrypts their traffic then the eavesdropper could not decode anything from the cipher-text. 2). If the model is fine-tuned on all parameters except the word embedding layer then the adversary could not get the bag-of-words in the first place.
3. In Section 4.2, the author mentioned that they could not determine the frequency of words. I don’t think this claim holds as one could learn frequency information from the norm of word embedding gradients, i.e. more frequent words tend to have seen more SGD updates and thus larger norms.
4. The precision number in Table 3 seems to suggest that recovering multiple sentences would be hard with FILM and results in quite a number of false positives.

---

> ### Author Response · Authors · 2022-08-02
> **Response to Reviewer FFKN**
>
> We thank the reviewer for the helpful feedback. We have tried to address all your comments in the following. Please let us know if you feel we haven’t fully addressed your comments. We will be happy to address them further.
>
> # Q1.Evaluation for defenses?
> Please refer to [response to common question #4](https://openreview.net/forum?id=dqgzfhHd2-&noteId=PU9MOie6ywKn).
>
> # Q2. Encryption as a defense?
> > If the server and client encrypts their traffic then the eavesdropper could not decode anything from the cipher-text.
>
> We agree that encrypting traffic through traditional cryptographic schemes in conjunction with certificate checks can help mitigate risks of a man-in-the-middle style attack. However, our attack also works in scenarios where the server wishes to recover a client’s private data, where the server would have full access to the gradients after decryption. Homomorphic encryption schemes (e.g. Nvidia Clara) can mitigate attacks in this scenario, but incur a performance penalty during training and are not widely adopted.
>
> We hope our work will motivate research and adoption of homomorphic encryption schemes for federated autoregressive language modeling, amongst other defenses.
>
> # Q3. Freezing embedding layers  as a defense?
> Please refer to [response to common question #2](https://openreview.net/forum?id=dqgzfhHd2-&noteId=necwm689O6D).
>
> # Q4. Recovering the frequency of tokens?
> Please refer to [response to common question #3](https://openreview.net/forum?id=dqgzfhHd2-&noteId=necwm689O6D).
>
> # Q5. High false-positive rates when recovering multiple sentences?
> > The precision number in Table 3 seems to suggest that recovering multiple sentences would be hard with FILM and results in quite a number of false positives. any thoughts?
>
> We agree that there are many false-positives (i.e. low precision) when recovering sentences in a batch when generating sentences over many iterations. However, for a smaller number of iterations (e.g. 10), we note that the recall is on average around 0.19, while precision is fairly high at an average of 0.28. This means that, in practice, 28% (i.e. a little less than 1 in 3) of the sentences an attacker generates over 10 iterations contain some information about the client’s private data. Certainly this should be considered a severe leakage of information.
> A clever attacker could potentially develop better heuristics to disambiguate true from false positives, given some other knowledge (e.g. knowledge of the world from a public dataset to rule out reconstructions with clearly incorrect statements), or by comparing reconstructions from multiple batches which share one or two of the same sentences (e.g. from different epochs of training); In this scenario, the intersection of the sets of reconstructions could further narrow the true positives from the false positives. This could be an interesting point of investigation for future papers in this direction - we hope our results motivate such study.
>
> # Q6. Performance with a malicious server?
> > Another common threat model is to have a malicious server who knows only about the gradients aggregated on multiple users on multiple batches. How would FILM work in such a threat model?
>
> Thanks for the insightful comment. Our current attack only assumes an honest-but-curious attacker for the sake of generalizability, but we agree with the reviewer that a malicious server is another interesting threat model.
>
> Ideally, FILM will be stronger when the server is malicious: as discussed in [1, 2], a malicious server can modify the model’s weights such that it can extract gradients for a single sample (i.e. sentence) from the batched gradients; In our case, such a malicious server can then use the per-sentence gradients as another supervision in the attack, which could potentially improve the attack performance. We plan to also include some ablation study for this threat model in the updated version.
>
> # Q7. Moving justification for sentence reordering earlier
> > I would appreciate it if authors could provide an intuition behind using gradient norm in the scoring function in Section 4.4 instead of appendix.
>
> Thanks for the suggestion. We will move the explanation to the main paper in the updated version (we apologize for not being able to update it now due to space limit).
>
> # Q8. Evaluation for limitations
> > I think the paper can be improved if there were quantitative measurements on the limitations.
>
> Thanks for the suggestion. We have added evaluation of several defenses (see common question #4); we plan to address other limitations in future work.
>
> # References
>
> [1] Boenisch, F., Dziedzic, A., Schuster, R., Shamsabadi, A. S., Shumailov, I., and Papernot, N. When the curious abandon honesty: Federated learning is not private.
>
> [2] Fowl, L., Geiping, J., Reich, S., Wen, Y., Czaja, W., Goldblum, M. and Goldstein, T., 2022. Decepticons: Corrupted transformers breach privacy in federated learning for language models. arXiv preprint.

---

> ### Author Response · Authors · 2022-08-08
> **Thanks again and following up**
>
> Dear Reviewer FFKN,
>
> Thank you again for your thoughtful review. We are very thankful for your comments and suggestions that helped improve our paper and have reflected them in our revision.
>
> As the end of the discussion is approaching, it will be great if we can learn whether our responses have addressed your concerns. We would also appreciate the opportunity to engage further if needed.
>
> Kind regards,
>
> Authors of Paper10606

---

### Official Review · Reviewer_x95N · 2022-07-11

**Rating:** 6
**Confidence:** 3
**Soundness:** 2 fair
**Presentation:** 3 good
**Contribution:** 3 good

**Summary:**

In this work, the authors present a model inversion attack for language models in the context of federated learning, termed FILM. A white-box attack scenario is considered, where an adversary has access to the global model computed by the central server and the local updates (gradients) of an (arbitrary) participating client. The proposed attack is carried out through three key pivotal steps. First, a bag of words is extracted through the non-zero rows of the client's embedding gradients. In the second step, sentences are reconstructed through a beam search algorithm. During the third, and final step, tokens appearing in the extracted sentence are ordered using a scoring mechanism that takes into account sentences' perplexity and gradient norm. Overall, this is an interesting and well-motivated work, with a clear structure, background, and contributions. However, elaboration is needed in the final empirical evaluation that is critical for replicating the presented results in federated settings and improving results completeness.

**Questions:**

W1: Why a larger batch size is important in Language Model (LM) tasks? It would be great if more background is given on the importance of the larger batch size. Moreover, does the attacker have full access to the entire vocabulary/embedding matrix during the white-box attack? It would be better to test the susceptibility of local datasets on a per-learner basis. For instance, what is the vulnerability (private text leakage) for each client? Do different data distributions (e.g., unbalanced, IID, or non-IID) affect the effectiveness of the attack? Is the attack method equally efficient across all clients' partitions? Moreover, what are some other federated domains that the proposed attack can be applied out of the box?

W2: Even though this work is placed in the federated setting, it is written as if the attack is tailored for centralized settings. Please elaborate on the federated environments you have evaluated. How many clients have you considered? How were the data/sentences distributed across clients? Are the data distributions easy to replicate so that future research can build on top and construct defensive mechanisms? Also, how many federation rounds did you train the models for? What are the update frequency and the aggregation method of local models? Training iterations should be expressed in terms of federation rounds (e.g., ref.2 below). To demonstrate model vulnerability due to overfitting another dimension would be attack success (at every federation round) versus the average training loss of the global model across all clients' local datasets.


Suggestions / Text Edits:
- missing $\theta$ from prob. distribution in equation 1
- line 69, 89: remove extra space
- line 71, 76: change work to works
- line 134: iterations to iteration
- Table 3 shows the standard deviation for precision and recall but the number of conducted trials is missing
- line 20, 61: consider adding the following references as well:

	ref.1: https://www.ndss-symposium.org/wp-content/uploads/2020/04/diss2020-23004-paper.pdf

	ref.2: https://proceedings.mlr.press/v143/gupta21a/gupta21a.pdf

**Limitations:**

The authors have discussed proposed approach's limitations.

**Strengths And Weaknesses:**

Strengths:
1. Well-written paper, easy-to-follow with clear motivation and contributions.
2. Interesting language model inversion attack that challenges existing language models training approaches.
3. Empirical results demonstrate the effectiveness of the attack in challenging natural language tasks.

Weaknesses:
1. Some details are missing from the attack setting.
2. Further discussion is needed in the evaluation section.

---

> ### Author Response · Authors · 2022-08-02
> **Response to Reviewer x95N (1/2)**
>
> We thank the reviewer for the helpful feedback. We have tried to address all your comments in the following. Please let us know if you feel we haven’t fully addressed your comments. We will be happy to address them further.
>
> # Q1. Why highlight the large batch size?
> > Why a larger batch size is important in Language Model (LM) tasks? It would be great if more background is given on the importance of the larger batch size.
>
> Language Model training usually uses a large batch size to save the number of total training time. Usually the batch size is increased according to the number of parameters, for instance, GPT-2 [1] with 117M parameters uses a batch size of 512, while GPT-3 [2] with 175B parameters uses a batch size of 3.2M - even the federated learning scheme acquires 10k active clients in each update, the batch size of each client will be as large as 320.
>
> Besides, from the attacker's perspective, the reconstruction becomes harder when batch size increases, as the attacker only has access to the averaged gradients of the batch. Therefore, designing attacks that can scale to larger batch sizes has been the focus of several previous works [3,4] and is considered as an important contribution.
>
> # Q2. Questions about the threat model
> > Moreover, does the attacker have full access to the entire vocabulary/embedding matrix during the white-box attack?
>
> Yes, we assume that the attacker has full access (i.e., white-box access) to the entire vocabulary/embedding matrix, as it is often regarded as public information about the language model being used; here is an [example](https://github.com/huggingface/transformers/issues/1458) for getting the embedding matrix for GPT-2. We’ve highlight this in our updated manuscript (see line 136).
>
> # Q3. Details for the federated environment
> > Please elaborate on the federated environments you have evaluated. How many clients have you considered? How were the data/sentences distributed across clients?
>
> Following previous studies [3, 4, 5, 6] for gradient inversion attacks in federated learning, our evaluation considers a setting with a single server and a single client. This setting is often accepted as an adequate stand-in for federated learning as synchronous federated learning with N clients, each with B samples per batch, is (assuming I.I.D. data) functionally equivalent to training a model with a batch size of N*B.
> > It would be better to test the susceptibility of local datasets on a per-learner basis. For instance, what is the vulnerability (private text leakage) for each client? Do different data distributions (e.g., unbalanced, IID, or non-IID) affect the effectiveness of the attack? Is the attack method equally efficient across all clients' partitions?
>
> Our setting of a single server and a single client naturally yields the same vulnerability for each client (as we only have one client). However, we agree that our work leaves open the question of non-I.I.D. data, which is a more realistic setting for federated learning. We are willing to investigate the performance of our attack under different federated learning configurations, such as varying the clients’ data distribution, federation rounds, or update frequency, as suggested by the reviewer.
>
> > Also, how many federation rounds did you train the models for? What are the update frequency and the aggregation method of local models?
>
> We train the model with early stopping to avoid overfitting; the training terminates at 90,000 and 15,000 federated rounds, respectively, for the WikiText-103 and Enron Emails datasets.

---

> > ### Author Response · Authors · 2022-08-02
> > **Responses to Reviewer x95N Cotd. (2/2)**
> >
> > # Q4. Reproducing results?
> > > Are the data distributions easy to replicate so that future research can build on top and construct defensive mechanisms?
> >
> > We’ve submitted our source code in the supplementary materials for reproduction. We also plan to open-source our implementation in the future. Please also refer to [common question #4](https://openreview.net/forum?id=dqgzfhHd2-&noteId=PU9MOie6ywKn) for evaluation of some existing defenses.
> >
> > # Minor comments:
> > - Remove extra space in line 69, 89
> > - Fixed typos in line 71, 76, 134
> > - Added the number of conducted trials to Fig 4
> > - Added suggested references to line 20, 61
> >
> > # References
> >
> > [1] Radford, A., Wu, J., Child, R., Luan, D., Amodei, D., Sutskever, I., et al. Language models are unsupervised multitask learners. OpenAI blog 2019.
> >
> > [2] Brown, T., Mann, B., Ryder, N., Subbiah, M., Kaplan, J.D., Dhariwal, P., Neelakantan, A., Shyam, P., Sastry, G., Askell, A. and Agarwal, S.. Language models are few-shot learners. NeurIPS 2020
> >
> > [3] Zhu, L., Liu, Z., and Han, S. Deep leakage from gradients. NeurIPS 2019.
> >
> > [4] Geiping, J., Bauermeister, H., Dröge, H., and Moeller, M. Inverting gradients–how easy is it to break privacy in federated learning? NeurIPS 2020.
> >
> > [5] Yin, H., Mallya, A., Vahdat, A., Alvarez, J. M., Kautz, J., and Molchanov, P. See through gradients: Image batch recovery via gradinversion. CVPR 2021.
> >
> > [6] Huang, Y., Gupta, S., Song, Z., Li, K., and Arora, S. Evaluating gradient inversion attacks and defenses in federated learning. NeurIPS 2021.

---

> ### Author Response · Authors · 2022-08-08
> **Thanks again and following up**
>
> Dear Reviewer x95N,
>
> Thank you again for your thoughtful review. We are very thankful for your comments and suggestions that helped improve our paper and have reflected them in our revision.
>
> As the end of the discussion is approaching, it will be great if we can learn whether our responses have addressed your concerns. We would also appreciate the opportunity to engage further if needed.
>
> Kind regards,
>
> Authors of Paper10606

---

> ### Comment · Reviewer_x95N · 2022-08-10
> **After Rebuttal**
>
> Here, I will provide a reply to my individual comments rather than to the collective response of the authors. First of all, I would like to thank the authors for the detailed response and for addressing all of my comments and for open-sourcing their code.
>
> However, after the authors' elaboration, I have some concerns regarding the investigated federated environments. A federated environment consisting of a single client is limited for empirical evaluation in federated settings. It would be better if the authors could provide some more results on the evaluation/effectiveness of the attack over an increasing number of clients and/or heterogeneous data distributions. Such results would make the work more compelling and more appropriate for federated settings since it could lead to more interesting (research) outcomes.

---

### Official Review · Reviewer_LfUD · 2022-07-11

**Rating:** 6
**Confidence:** 5
**Soundness:** 3 good
**Presentation:** 4 excellent
**Contribution:** 3 good

**Summary:**

The submission titled "Recovering Private Text in Federated Learning of Language Models" discusses privacy attacks in federated learning that recover private information from user gradients. The proposed attack first recovers a bag of words from the user update and then applies a series of beam searches to uncover sentences of user data. The approach is validated for a GPT-2 model with a batch size of up to 128 and sentences from enron and wikitext103.

**Questions:**

Some questions for the author response:

* Is it not clear to me why this submission restricts the attack to only single sentences. What happens when the full range of 1024 tokens in GPT is filled by a user update, instead of only short sentences? Relatedly, what is the average sequence length evaluated in the experimental section?

* Another question I have is concerning memorization. How many epochs are each of these models finetuned? Is it possible that the enron example leaks more data because every example is seen more often during training? Would it be possible to mark on Fig.4 when the data is first repeated?

* The proposed approach seems limited to autoregressive models to my understanding? This would be ok, but I wonder if the authors comment on the feasibility of using this attack on a masked language model?



Other minor remarks:
* There is a typo in line 285.

* The panels in Fig.5 are flipped (or at least referenced in text in the opposite way?)

* I further think this submission could be improved by discussing Dimitrov et al. in a bit more detail in the related work, given that both approaches use language priors and gradient information, just in different stages of the recovery (but I do not think a comparison is necessary).

* Minor: The submission discusses the oracle case where word frequencies are known to the attacker, and generally approximates word frequencies using the procedure proposed in Sec.4. However, word frequencies can also be recovered from the user gradients in addition to the basic bag-of-words by applying the approach of Wainakh et al, "User Label Leakage from Gradients in Federated Learning". This is also described in Fowl et al., " Decepticons: Corrupted transformers breach privacy in federated learning for language models". I wonder if this would be helpful to improve the attack?

**Limitations:**

The submissions discusses some limitations and ethical considerations and does so adequately.

**Strengths And Weaknesses:**

Overall I like the general approach proposed by the authors. The idea to re-use the language model itself to score its sentences and guide towards sentences likely contained in the gradient update is sensible and executed well. The submission features a range of interesting case studies, especially considering partial memorization and intermediate model states that I especially like. I do have a few smaller concerns regarding experimental evaluation,  ablation studies and related work that I would like to discuss:

* The ROUGE scores in Sec.5 are computed by comparing the recovered sentence to all ground truth sentences as possible references if  my understanding is correct. I wonder whether this measure is overestimating the effectiveness of the attack as the number of references increases. It would be good to include baselines of a) a random sentence generated by beam search from the current model and b) a sentence generated only from beam-search starting from the bag-of-words (so only the first step of the proposed search). I also wonder what the peak accuracy would be, i.e. comparing the recovered sentence to all reference sentences in direct accuracy and choosing the highest accuracy.

* Relatedly, this submission could be greatly improved with an ablation study of the sequence of improvements discussed in Sec.4, which I could not find in the appendix.

* This work mentions related work in Deng et al. "Tag: Gradient attack on transformer-based language models." and Dimitrov et al., "Lamp: Extracting text from  gradients with language model priors", but does not return to comparisons to these more recent works in the experimental evaluation. While this is (to me) understandable for the case of Dimitrov et al., given the complexity of the attack therein and the recency of that paper, but implementing Deng et al. should only be a minor change to the already present evaluation of the Zhu et al. baseline. In Dimitrov et al., the attack of Deng et al. was able to noticeably increase the baseline effectiveness of this type of matching attack.


Overall, these are the reasons why I currenly only score this work as a weak accept, but I am interested in discussing these and increasing my score in the future.

---

> ### Author Response · Authors · 2022-08-02
> **Response to Reviewer LfUD (1/2)**
>
> We thank the reviewer for the helpful feedback. We have tried to address all your comments in the following. Please let us know if you feel we haven’t fully addressed your comments. We will be happy to address them further.
>
> # Q1. Justification for the ROUGE score
> > I wonder whether this measure is overestimating the effectiveness of the attack as the number of references increases. It would be good to include baselines of a) a random sentence generated by beam search from the current model and b) a sentence generated only from beam-search starting from the bag-of-words (so only the first step of the proposed search).
>
> Thanks for the comment. We have added a comparison of ROUGE-L scores in a scenario where the bag of words was not used by an attacker during beam search. We compare the ROUGE-L scores from both scenarios for a batchsize of 1, given the same word prompts. There is a clear difference in performance between beam search with a bag of words and without a bag of words.
>
> We do however note that as batch sizes grow larger, the relative improvement by using the bag-of-words decreases. This aligns with intuition - an attacker without access to a bag of words can be thought of as having a bag of all possible words in the dictionary. As the batch size grows, the bag of words grows as well - approaching the scenario where the bag consists of the entire dictionary.
>
> > I also wonder what the peak accuracy would be, i.e. comparing the recovered sentence to all reference sentences in direct accuracy and choosing the highest accuracy.
>
> We have updated our manuscript with a new figure (see: Figure 7 in Appendix C.2) which demonstrates the upper bounds on recovery performance for each batch size and dataset.  We found only a single sentence in Wikitext-103, and 3 sentences in Enron Emails (both out of samples of 20) which were perfectly recovered. Perfect recoveries were only found for batch sizes of 1. However, even for larger batch sizes, we observe a high peak accuracy (i.e. >.5 ROUGE-L score) - indicating that recovery of the majority of a sentence is still possible even in this setting.
>
> # Q2. Ablation study of the sequence of improvements
> > this submission could be greatly improved with an ablation study of the sequence of improvements discussed in Sec.4, which I could not find in the appendix.
>
> Thanks for the suggestion. Here we demonstrate the sequence of improvements (we plan to make this clearer in the final version of the manuscript):
> - In Figure 10 (Appendix C.7): Overall we find that beam search is significantly stronger for small batchsizes if the attacker also has a bag of words. Moreover, using the bag of words allows attackers to reconstruct significantly more named entities even in larger batch sizes.
> - In Figure 6: the difference between the “after beam search” curve and the “after sentence reordering” curve shows the reconstruction performance after beam search (Section 4.3), and after sentence reordering (Section 4.4). Perhaps the dashed line style makes it hard for readers to notice, so we’ve updated the figure by switching to another line style.
>
> # Q3. Comparison with TAG
> Please refer to [response to common question #1](https://openreview.net/forum?id=dqgzfhHd2-&noteId=smNM_fTegn).
>
> # Q4. Restriction to single sentence evaluation
> > Is it not clear to me why this submission restricts the attack to only single sentences. What happens when the full range of 1024 tokens in GPT is filled by a user update, instead of only short sentences?
>
> Thanks for the comment. We believe recovering a single sentence already reveals serious privacy concerns. We agree that it is challenging to recover even longer sequences: ideally, the attack will be weakened if the full range of 1024 tokens in GPT is filled, due to the increased search space induced by a much larger bag of tokens. We plan to report this as an ablation study in the updated version as well.
>
> > Relatedly, what is the average sequence length evaluated in the experimental section?
>
> The average/std sequence length evaluated is 24.05/10.85 for the WikiText dataset, and 20.09/6.88 for the Enron email dataset. For reference, the average sentence length in SST-2 is 19.

---

> > ### Author Response · Authors · 2022-08-02
> > **Response to Reviewer LfUD Cotd. (2/2)**
> >
> > # Q5. Experiment details
> > > How many epochs are each of these models finetuned? Is it possible that the enron example leaks more data because every example is seen more often during training?
> >
> > Figure 4 tracks the number of iterations of training - each iteration is training on a batch of 64 sentences (the largest power of two before we experience out-of-memory issues on our hardware). As the dataset size for wikitex-103 is 203,456 sentences, the model repeats sentences every 203,456/64 = 3179 iterations. We note that memorization by the model is already very strong by about 10,000 iterations, which means the model has seen every sentence around 3 times. Notably, the model does not stop training (i.e. experience an inflection in test loss) until around 90,000 iterations (i.e. 30 repetitions).
> >
> > For the Enron Emails dataset, the model repeats sentences every 30,000/64 = 468 iterations. Training in Enron terminates at around 15,000 iterations (i.e. 32 repetitions).
> >
> > Previous work [4] indicates that memorization can occur even when training under a small number of epochs.
> >
> > > Would it be possible to mark on Fig.4 when the data is first repeated?
> >
> > We have marked the 12716th interactions on Fig.4 in the updated manuscript.
> >
> > # Q6. More discussion for the LAMP attack
> > > This submission could be improved by discussing Dimitrov et al. in a bit more detail in the related work.
> >
> > Thanks for the suggestion. We’ve discussed the LAMP paper (Dimitrov et al.) with more technical details in the updated version (see lines 80 - 83).
> >
> > # Q7. Generalizing to the masked language model?
> > > The proposed approach seems limited to autoregressive models to my understanding? This would be ok, but I wonder if the authors comment on the feasibility of using this attack on a masked language model?
> >
> > We agree that our method is restricted to autoregressive models - the core limitation here is in the beam search to discover sequences of memorized tokens. While autoregressive models are well-suited for beam search as text is generated sequentially, it is not immediately obvious how this approach would translate to a masked language model. Previous work [1,2,3] identifies that there is some degree of memorization of training data in BERT models trained on medical datasets, which is recoverable using alternate sampling methods such as nucleus sampling and top-k sampling. All of these prior approaches would certainly benefit from having access to a bag of words to restrict the search space.
> >
> > # Minor comments:
> > - Fixed the typo in line 285.
> > - Corrected the reference for Figure 5 in text
> >
> > # References
> >
> > [1] E. Lehman, S. Jain, K. Pichotta, Y. Goldberg, and B. C. Wallace, “Does BERT Pretrained on Clinical Notes Reveal Sensitive Data?” arXiv, Apr. 22, 2021. Accessed: Aug. 01, 2022. [Online]. Available: http://arxiv.org/abs/2104.07762
> >
> > [2] T. Vakili and H. Dalianis, “Are Clinical BERT Models Privacy Preserving? The Difficulty of Extracting Patient-Condition Associations,” p. 7.
> >
> > [3] A. Wang and K. Cho, “BERT has a Mouth, and It Must Speak: BERT as a Markov Random Field Language Model,” p. 7.
> >
> > [4] N. Carlini et al., “Extracting Training Data from Large Language Models,” arXiv:2012.07805 [cs], Jun. 2021, Accessed: Oct. 22, 2021. [Online]. Available: http://arxiv.org/abs/2012.07805

---

> > > ### Comment · Reviewer_LfUD · 2022-08-05
> > > **Response**
> > >
> > > Thank you providing these extensive experiments and ablation studies, especially concerning the memorization effects. There is one point where I am slightly more confused than before - reading your response and the sections describing the updated Fig.4: How is the data actually split into samples on which the attack is evaluated vs samples on which the model is finetuned? I could only find the description for the finetuning part.

---

> > > > ### Author Response · Authors · 2022-08-06
> > > > **Response to follow-up question**
> > > >
> > > > Thank you for your response and for the follow-up question.
> > > >
> > > > > How is the data actually split into samples on which the attack is evaluated vs samples on which the model is finetuned?
> > > >
> > > > To clarify, *"samples on which the attack is evaluated"* and *"samples on which the model is finetuned"* are not disjoint - as described in line 252 of our manuscript, the attack is evaluated with **a subset of** the dataset on which the model is fine-tuned. This is because the goal of the attacker is to recover private training (i.e. fine-tuning) data batch from gradients in federated training of the language modeling task (see "Adversary’s objective" in Section 3.3).
> > > >
> > > > Please let us know if you feel we haven’t fully addressed your concerns. We will be happy to address them further.

---

> > > > > ### Comment · Reviewer_LfUD · 2022-08-08
> > > > > **Thanks**
> > > > >
> > > > > Thank you for the follow-up clarifications.
> > > > >
> > > > > For the record, the investigated scenario is a bit of a departure to previous work (e.g. in DLG/TAG, maybe also LAMP?), which was concerned with "1-shot recovery" of data so far unseen during training, as would occur in cross-device federated learning. This makes comparisons to previous work a bit less relevant, as previous does not utilize the additional information that the model has memorized the training data. That being said, I do agree that this is a worthwhile setting to investigate, closely related to cross-silo scenarios, and do not see this as a concern.

---

> > > > > > ### Author Response · Authors · 2022-08-08
> > > > > > **Response**
> > > > > >
> > > > > > We thank the reviewer for their response - indeed, we agree that DLG/TAG/LAMP focus on 1-shot recovery. We are also glad that the reviewer sees the value in investigating different settings.
> > > > > >
> > > > > > We would like to clarify that our method still works in a cross-device federated learning setting. An attacker using DLG/TAG/LAMP would require access to exactly the same information as our approach - **namely, the parameters of the model and the gradients at some training iteration $t$**. The key distinctions are:
> > > > > >
> > > > > > - **Methodology**: Our approach uses the same information in a different way (prioritizing recovery of memorized data directly from model parameters, aided by embeddings) instead of directly from gradients. Moreover, we would like to stress that despite reliance on the same information, existing defenses such as gradient pruning do not protect against our attack.
> > > > > > - **Performance**: While DLG/TAG/LAMP’s performance is likely to be independent of $t$, our attack usually yields stronger performance with a larger $t$ because of memorization of the training data (see Figure 4 in our manuscript).
> > > > > >
> > > > > > Please let us know if there are any additional questions we can address. Thanks!

---

### Author Response · Authors · 2022-08-02
**Thanks for your valuable feedback and a summary of comments (1/4)**

We thank AC and all reviewers for their time and valuable feedback, and for the recognition that our manuscript "is an interesting and well-motivated work", and "features a range of interesting case studies".

We summarize common concerns and respond to them below. We also summarize [a list of individual concerns](https://openreview.net/forum?id=dqgzfhHd2-&noteId=Ab0VPfZwQa), and will respond to them in posts to individual reviewers.

# Q1. How does the proposed attack compare to previous works (Reviewers LfUD & vs2F)?
> Results in Table 1 show that DLG and TAG essentially can't reconstruct anything even for batch size 1. Moreover, authors claim that TAG doesn't improve over DLG (footnote in Table 1). This all clashes with results in Deng et al. and Dimitrov et al. where DLG has much better reconstruction than shown in this submission, and using TAG further improves the reconstructions.

Thanks for pointing this out.  The DLG results we reported are based on [the official implementation](https://github.com/mit-han-lab/dlg) but the results are poor as we demonstrated. As TAG is not open-sourced, we were unable to fully verify our implementation and thus didn't include the results in the submission. We now instead evaluate the performance of TAG using a more recent [third-party implementation](https://github.com/JonasGeiping/breaching). By our updated testing, TAG moderately outperforms DLG across all batch sizes, and seems to be less sensitive to initial conditions - an improvement which we have updated in our manuscript (see Table 1 & Table 7). However, both DLG and TAG fail to recover the order of words in a sentence (as suggested by low ROUGE-2 & ROUGE-L scores).

We wish to also emphasize that the original DLG and TAG experiments consider recovery in masked language models under a batchsize of 1 - our approach instead focuses on recovering from autoregressive language models and under larger batch sizes.

**Table 1. Comparison of text reconstructions from gradients for various datasets, prior methods, and batch sizes (see Table 7 in Appendix C.1 for larger batch sizes). R-1, R-2, and R-L, denote average ROUGE-1, ROUGE-2 and ROUGE-L scores, respectively. All data represents average values collected from 20 samples.**
|  |  | Batch size = 1 |  |  |  | Batch size = 4 |  |  |  |
|:---:|:---:|:---:|:---:|:---:|:---:|:---:|:---:|:---:|:---:|
|  |  | R-1 | R-2 | R-L | NERR | R-1 | R-2 | R-L | NERR |
| Wikitext-103 | DLG | 0.20 | 0.02 |  0.15 | 0.09 | 0.17  | 0.01 | 0.14  | 0.07 |
|  | TAG | 0.38  | 0.06 | 0.27 |  0.00 | 0.32 | 0.07 | 0.23 | 0.18 |
|  | FILM (ours) | **0.74** | **0.44** | **0.54** | **0.25** | **0.48** | **0.23** | **0.37** | **0.26** |
| Enron email | DLG | 0.24 | 0.03 | 0.18 | 0.07 | 0.21 | 0.02 | 0.17 | 0.04 |
|  | TAG | 0.42  | 0.12 | 0.33 | 0.21 | 0.39 | 0.10 | 0.29 | 0.46 |
|  | FILM (ours) | **0.83** | **0.57** | **0.70** | **0.45** | **0.54** | **0.32** | **0.44** | **0.54** |

To further clarify the differences between our work and the prior work, we have created the following table:

**Table 2. Summary of data reconstruction from gradients in federated learning of NLP tasks.**
| **Name** | **Technique** | **Original Batchsize?** | **Model** | **Datasets (Sequence Length, Task)** |
|---|---|---|---|---|
| DLG | End-to-End optimization | Only tested for batchsize 1 | BERT | Masked Language Modeling (~30) |
| TAG | End-to-End optimization + Regularization term | Only tested for batchsize 1 | TinyBERT, BERT, BERTLARGE | CoLA (5-15, Sentence Classification), SST-2 (10-30, Sentiment Analysis), RTE (50-100, Textual Entailment) |
| LAMP | End-to-End optimization (with regularization) + Discrete token swapping | <=4 | TinyBERT, BERT, BERTLARGE | CoLA (5-9)  SST-2 (3-13), Rotten Tomatoes (14-27, Sentiment Analysis) |
| FILM (Ours) | Bag-of-words extraction + Discrete beam search + Discrete Reordering | <=128 | GPT-2 | Wikitext-103 (15-40), Enron Emails (15-40) (Both Autoregressive Language Modeling) |

Our method (FILM) differs greatly in setting from that of prior work, which only focuses on tasks in which the label is binary, and hence can be easily recovered using an end-to-end optimization technique. However, for a task such as autoregressive language modeling (our setting), recovery of labels is significantly more difficult, as the space of potential labels is larger. Our method instead does not rely on any end-to-end optimization at all. Moreover, our approach is able to recover sensitive information even from large batch sizes, whereas end-to-end methods do not scale as well to larger batch sizes (Appendix C.1).

We would like to additionally emphasize that FILM is an entirely new attack vector for federated language models, and hence warrants discussion about defenses and mitigations (e.g. frozen word embeddings) which are independent from those typically used to protect against end-to-end optimization attacks (e.g. gradient pruning).

---

> ### Author Response · Authors · 2022-08-02
> **Common Questions Cotd. (2/4)**
>
> # Q2. Is freezing the embedding layer a potential mitigation? (Reviewers FFKN & vs2F)?
> We agree that our attack method assumes that word embeddings need to be trained. However, we would like to point out that training word embeddings is a common practice in training language models, especially when we need to train language models from the scratch, or fine-tuning a language model on a new domain. Training word embeddings in language models is a standard practice especially in language modeling tasks.
>
> We additionally experimented with training a language model from scratch with frozen embeddings and found the final perplexity on the test set to be significantly worse (68.37 with frozen embeddings vs 15.96 with unfrozen embeddings).
>
> We hope that our attack highlights the risks associated with revealing embedding gradients to an attacker, and motivates further investigation into the tradeoffs made when freezing embedding gradients as a defense.
>
> # Q3. Is it possible to predict token  frequencies from embedding gradients (Reviewers FFKN & LfUD)?
> We’ve actually tried to train a regression model to predict token frequencies (following [1,2]), which takes as input 1) the norm of word embedding gradients (ideally, more frequent tokens tend to have larger norms), 2) a reference frequency (i.e., the token’s frequency in a very large corpus, e.g., Wikipedia or WikiText).
>
> * As shown in Table 3 (below), although we can estimate the frequency of words based on the magnitude of gradients (with an averaged prediction accuracy of >97%), we find very frequent words (>=95 percentile) are difficult to predict accurately, which is probably due to the fact that the label distribution is highly unbalanced for frequent and infrequent tokens.
>
> * **More importantly**, we find our approach can bypass this step by leveraging the n-gram penalty (see Section 4.3 for methodology): as shown in Figure 5 in our manuscript, the gap between frequency known and unknown is very small when we use the n-gram penalty with n=2. That means, using any frequency estimator won't further improve the performance much.
>
> We’ve updated our manuscript (see lines 175-178) to highlight both findings and plan to add these results into the final version of the manuscript.
>
> (For reference, here are some examples for frequent tokens (which usually appear three times or more in a batch): ‘a’, ‘an’, ‘and’, ‘are’, ‘be’, ‘by’ ‘for’, ‘is’, ‘the’; most of them are article, preposition or conjunction. An accurate estimation for the frequency of these words is very crucial to the beam search, as missing one of such words may totally change the structure of the sentence.)
>
> **Table 3. Prediction accuracy for token frequencies of different subgroups (batch size=16). The model predicts infrequent tokens much more accurately than frequent tokens.**
> | Percentile in token distribution | Token Frequency | Prediction accuracy |
> |---|---|---|
> | <90 | [1, 1] | 99.81% |
> | [90, 95) | [2, 3) | 93.26% |
> | [95, 99) | [3, 13) | 76.23% |
> | >= 99 | [13, 37] | 23.10% |
> | Average accuracy |  | 97.77% |

---

> > ### Author Response · Authors · 2022-08-02
> > **Common Questions Cotd. (3/4)**
> >
> > # Q4. Presenting the attack performance with defense applied?
> > As the set of tokens of the private batch already leakage a significant amount of information, we evaluate the performance of the first step of our attack pipeline (i.e. recovering the set of tokens from gradients) under two defenses: gradient pruning and differentially private stochastic gradient descent (DPSGD). For each defense, we propose an adaptive attack strategy and present the results,
> >
> > We plan to extend this study to the whole attack pipeline in the updated manuscript.
> >
> > **Datasets and metrics**: The evaluation uses the WikiText-103 dataset and the GPT-2 model (with batch size = 16). Given $S^*$, the recovered set of tokens by the attack,  and $S$, the original set of tokens in the private batch, we use the following two metrics to evaluate the performance of the attack:
> > * Precision: $|S^* \cap S| / |S^*|$, which is the fraction of original tokens in the recovered set.
> > * Recall:  $|S^* \cap S| / |S|$, which is the fraction of recovered original tokens in all original tokens.
> >
> > **Attacking gradient pruning**: Gradient pruning zeros out the fraction of gradient entries with low magnitude. However, the embedding gradients of existing tokens in the batch are still non-zero unless the prune ratio p is extremely high. Thus, the attack strategy remains the same as the vanilla attack: we retrieve the tokens whose embedding gradients are non-zero. As shown in Table 4 (below), the attack always returns existing tokens in the private batch (i.e. precision is always 1). The recall decreases as the prune ratio increases, because some entries of word embedding gradients get completely zeroed out and thus the corresponding tokens cannot be retrieved by the attack. However, the attacker can still recover a considerable amount of tokens (i.e. >90%) even with a prune ratio as high as 0.9999.
> >
> > **Table 4. Precision and recall for the reconstruction of tokens under the gradient pruning defense. Our attack can still recover a considerable amount of tokens (i.e. >90%) even with a prune ratio as high as 0.9999.**
> > | Prune ratio | Perplexity of the trained model  | Precision of token recovery | Recall of token recovery |
> > |---|---|---|---|
> > | 0 | 11.46 | 1 | 1 |
> > | 0.9 | 11.57 | 1 | 1 |
> > | 0.95 | 12.58 | 1 | 1 |
> > | 0.99 | 12.77 | 1 | 0.998 |
> > | 0.999 | 15.34 | 1 | 0.983 |
> > | 0.9999 | 19.21 | 1 | 0.902 |
> >
> > **Attacking DPSGD**: The strategy to launch the attack under DPSGD is more tricky, as the gradients become noisy and the previous heuristics of returning non-zero gradient entries no longer hold. We come up with a new attack strategy which involves using a threshold $\tau = \sigma \sqrt{2\log d}$ to discriminate noisy gradients with pure noise, where d is the embedding dimension (i.e. 768) and $\sigma$ is the noise scale of DPSGD. For each token, we check the maximum magnitude of its embedding gradients: if the value is larger than $\tau$, then the token may be included in the original batch with high probability.
> >
> > As shown in Table 5 (below), the attack performance drops when the epsilon of DPSGD scheme decreases (at the cost of perplexity), because the relative scale between noise magnitudes and original gradient magnitudes increases. However, the attack becomes easier for larger batch sizes,  because the gradients are summed over multiple samples and the noise becomes less dominant.
> >
> > **Table 5. Precision and recall for the reconstruction of tokens under the DPSGD defense. Our attack fails to retrieve the majority of tokens (i.e., recall < 0.5) when epsilon is equal to or smaller than 1. However, the performance drop measured by perplexity is around 6.4.**
> >
> > | $\epsilon$ of DP | Perplexity of the trained model  | Precision of token recovery | Recall of token recovery |
> > |---|---|---|---|
> > | 0.1 | 25.78 | 0.233 | 0.137 |
> > | 0.5 | 18.94 | 0.401 | 0.236 |
> > | 1 | 17.88 | 0.493 | 0.440 |
> > | 5 | 14.23 | 0.672 | 0.631 |
> > | inf | 11.46 | 1 | 1 |

---

> > > ### Author Response · Authors · 2022-08-02
> > > **Common Questions Cotd. (4/4)**
> > >
> > > # Individual comments
> > > We have summarized other comments and our responses as below (corresponding reviewer IDs are provided in parenthesis). Detailed responses can be found in posts to individual reviewers.
> > >
> > > 1. **Threat model**: We’ve clarified the threat model in our evaluation (reviewer x95N’s Q2), and discussed how the attack can generalize to an even stronger threat model that assumes a malicious server (reviewer FFKN’s Q3)
> > > 2. **Experimental setup**: We provided more details for our training configurations (reviewer LfUD’s Q3), federated setting (reviewer x95N’s Q3), and justified our choice of metric (reviewer LfUD’s Q1). We also would like to kindly point reviewers to our source code (in the submitted supplementary materials) for questions about reproduction (reviewer x95N’s Q4).
> > > 3. **Experimental results**: We interpreted our results for multi-sentence reconstruction  (reviewer FFKN Q2 & reviewer vs2F’s Q4), and provided analysis for of which sentence in the batch is usually reconstructed (reviewer vs2F’s Q4).
> > > 4. **Discussion of potential limitations**: we discussed the limitation of only presenting sentence-level evaluation (reviewer LfUD’s Q2)
> > > 5. We’ve also updated our manuscript (with changes highlighted in red) to include comparison with previous works (common question #1), discuss related attacks (reviewer LfUD’s  Q4) and defenses (reviewer FFKN’s  Q1) in more detail and fixed the typos.
> > >
> > > # References
> > > [1] Fowl, L., Geiping, J., Reich, S., Wen, Y., Czaja, W., Goldblum, M. and Goldstein, T., 2022. Decepticons: Corrupted transformers breach privacy in federated learning for language models. arXiv preprint arXiv:2201.12675.
> > >
> > > [2] Wainakh, A., Ventola, F., Müßig, T., Keim, J., Cordero, C.G., Zimmer, E., Grube, T., Kersting, K. and Mühlhäuser, M., 2021. User label leakage from gradients in federated learning. arXiv preprint arXiv:2105.09369.

---

### Meta-Review · Area_Chair_yCyz · 2022-08-27

**Recommendation:** Accept
**Confidence:** Less certain

**Metareview:**

This paper proposes a novel gradient inversion attack to recover private text in FL training under the honest-but-curious server threat model. Through extensive experimentation, the authors demonstrate that their attack achieves improved attack performance compared to prior work under the same threat model, and performed plenty of ablation studies to analyze the effect of batch size, stage of model training, hyperparameters, etc.

One major weakness is that the method requires the token embedding layer to be trained, whereas real-world applications often use pre-trained embeddings. The authors showed that their attack performance degrades significantly when the token embeddings are frozen. Other weaknesses include limitation of the attack to autoregressive models, and inadequate discussion regarding the honest-but-curious threat model and secure aggregation (reviewer FFKN).

After the discussion phase, most reviewers agreed that the above weaknesses are minor and that the paper’s contribution warrants acceptance. AC therefore recommends acceptance to NeurIPS, but strongly encourages the authors to revise their draft to address concerns regarding frozen embedding—by performing more extensive experiments to demonstrate the method’s potential limitations, and more clearly position their attack under the setting of an honest-but-curious server with/without secure aggregation.


**Award:**

No

---

### Decision · Program_Chairs · 2022-09-14

Accept